# e-COP : Episodic Constrained Optimization of Policies

**Akhil Agnihotri**
University of Southern California
agnihotri.akhil@gmail.com

**Rahul Jain**
Google DeepMind and USC
rahulajain@google.com

**Deepak Ramachandran**
Google DeepMind
ramachandrand@google.com

**Sahil Singla**
Google DeepMind
sasingla@google.com

## Abstract

In this paper, we present the `e-COP` algorithm, the first policy optimization algorithm for constrained Reinforcement Learning (RL) in episodic (finite horizon) settings. Such formulations are applicable when there are separate sets of optimization criteria and constraints on a system's behavior. We approach this problem by first establishing a policy difference lemma for the episodic setting, which provides the theoretical foundation for the algorithm. Then, we propose to combine a set of established and novel solution ideas to yield the `e-COP` algorithm that is easy to implement and numerically stable, and provide a theoretical guarantee on optimality under certain scaling assumptions. Through extensive empirical analysis using benchmarks in the Safety Gym suite, we show that our algorithm has similar or better performance than SoTA (non-episodic) algorithms adapted for the episodic setting. The scalability of the algorithm opens the door to its application in safety-constrained Reinforcement Learning from Human Feedback for Large Language or Diffusion Models.

## 1 Introduction

RL problems may be formulated in order to satisfy multiple simultaneous objectives. These can include performance objectives that we want to maximize, and physical, operational or other objectives that we wish to constrain rather than maximize. For example, in robotics, we often want to optimize a task completion objective while obeying physical safety constraints. Similarly, in generative AI models, we want to optimize for human preferences while ensuring that the output generations remain safe (expressed perhaps as a threshold on an automatic safety score that penalizes violent or other undesirable content). Scalable policy optimization algorithms such as TRPO [31], PPO [33], etc have been central to the achievements of RL over the last decade [34, 38, 7]. In particular, these have found utility in generative models, e.g., in the training of Large Language Models (LLMs) to be aligned to human preferences through the RL with Human Feedback (RLHF) paradigm [2]. But these algorithms are designed primarily for the unconstrained infinite-horizon discounted setting: Their use for constrained problems via optimization of the Lagrangian often gives unsatisfactory constraint satisfaction results. This has prompted development of a number of constrained policy optimization algorithms for the infinite-horizon discounted setting [3, 39, 43, 41, 19, 6, 5], and for the average setting [4].

However, many RL problems are more accurately formulated as episodic, i.e., having a finite time horizon. For instance, in image diffusion models [10, 21], the denoising sequence is really a *finite* step trajectory optimization problem, better suited to be solved via RL algorithms for the episodic setting. When existing algorithms for infinite horizon discounted setting are used for such problem, they exhibit sub-optimal performance or fail to satisfy task-specific constraints by prioritizing short-term constraint satisfaction over episodic goals [11, 28, 17]. Furthermore, the episodic setting allows for objective functions to be time-dependent which the infinite-horizon formulations do not. Even when the objective functions are

time-invariant, there is a key difference: for non-episodic settings, a stationary policy that is optimal exists whereas for episodic settings, the optimal policy is always non-stationary and time-dependent. This necessitates development of policy optimization algorithms specifically for the episodic constrained setting. We note that such policy optimization algorithms do not exist even for the unconstrained episodic setting.

In this paper, we introduce e-COP, a policy optimization algorithm for episodic constrained RL problems. Inspired by PPO, it uses deep learning-based function approximation, a KL divergence-based proximal trust region and gradient clipping, along with several novel ideas specifically for the finite horizon episodic and constrained setting. The algorithm is based on a technical lemma (Lemma 3.1) on the difference between two policies, which leads to a new loss function for the episodic setting. We also introduce other ideas that obviate the need for matrix inversion thus improving scalability and numerical stability. The resulting algorithm improves performance over state-of-the-art baselines (after adapting them for the episodic setting). In sum, the e-COP algorithm has the following advantages: (i) *Solution equivalence:* We show that the solution set of our e-COP loss function is same as that of the original CMDP problem, leading to precise cost control during training and avoidance of approximation errors (see Theorem 3.3); (ii) *Stable convergence:* The e-COP algorithm converges tightly to safe optimal policies without the oscillatory behavior seen in other algorithms like PDO [13] and FOCOPS [43]; (iii) *Easy integration:* e-COP follows the skeleton structure of PPO-style algorithms using clipping instead of steady state distribution approximation, and hence can be easily integrated into existing RL pipelines; and (iv) *Empiricial performance:* the e-COP algorithm demonstrates superior performance on several benchmark problems for constrained optimal control compared to several state-of-the-art baselines.

**Our Contributions and Novelty.** We introduce the first policy optimization algorithm for episodic RL problems, both with and without constraints (no constraints is a special case). While some of the policy optimization algorithms can be adapted for the constrained setting via a Lagrangian formulation, as we show, they don't work so well empirically in terms of constraint satisfaction and optimality. The algorithm is based on a policy difference (technical) lemma, which is novel. We have gotten rid of Hessian matrix inversion, a common feature of policy optimization algorithms (see, for example, PPO [33], CPO [3], P3O [41], etc.) and replaced it with a quadratic penalty term which also improves numerical stability near the edge of the constraint set boundary - a problem unique to constrained RL problems. We provide an instance-dependent hyperparameter tuning routine that generalizes to various testing scenarios. And finally, our extensive empirical results against an extensive suite of baseline algorithms (e.g., adapted PPO [33], FOCOPS [43], CPO [3], PCPO [39], and P3O [41]) show that e-COP performs the best or near-best on a range of Safety Gym [12] benchmark problems.

**Related Work.** A broad view of planning and model-free RL techniques for Constrained MDPs is provided in [27] and [18]. The development of SOTA policy optimization started with the TRPO algorithm [31], which used a trust region to update the current policy, and was further improved in PPO by use of proximal ideas [33]. This led to works like CPO [3], RCPO [36], and PCPO [39] for constrained RL problems in the infinite-horizon discounted setting. ACPO [4] extended CPO to the infinite-horizon average setting. These methods typically require inversion of a computationally-expensive Fischer information matrix at each update step, thus limiting scalability. Lagrangian-based algorithms [29, 30] showed that you could incorporate constraints but constrained satisfaction remained a concern. Algorithms like PDO [13] and RCPO [36] also use Lagrangian duality but to solve risk-constrained RL problems and suffer from computational overhead. Some other notable algorithms include IPO [26], P3O [41], APPO [15], etc. that use penalty terms, and hence do not suffer from computational overhead but have other drawbacks. For example, IPO assumes feasible intermediate iterations, which cannot be fulfilled in practice, P3O requires arbitrarily large penalty factors for feasibility which can lead to significant estimation errors. We note that all the above algorithms are for the infinite-horizon discounted (non-episodic) setting (except ACPO [4], which is for the average setting). We are not aware of any policy optimization algorithm for the episodic RL problem, with or without constraints.

## 2   Preliminaries

An episodic, or fixed horizon Markov decision process (MDP) is a tuple, $\mathcal{M} := (S, A, r, P, \mu, H)$, where $S$ is the set of states, $A$ is the set of actions, $r : S \times A \times S \to \mathbb{R}$ is the reward function, $P : S \times A \times S \to [0, 1]$ is the transition probability function such that $P(s'|s, a)$ is the probability of transitioning to state $s'$ from state $s$ by taking action $a$, $\mu : S \to [0, 1]$ is the initial state distribution, and $H$ is the time horizon for each episode (characterized by a terminal state $s_H$).

A policy $\pi : S \to \Delta(A)$ is a mapping from states to probability distributions over the actions, with $\pi(a|s)$ denoting the probability of selecting action $a$ in state $s$, and $\Delta(A)$ is the probability simplex over the action space $A$. However, due to the temporal nature of episodic RL, the optimal policies are generally not stationary, and we index the policy at time $h$ by $\pi_h$, and denote $\boldsymbol{\pi}_{1:H} = (\pi_h)_{h=1}^H$. Then, the total undiscounted reward objective within an episode is defined as

$$ J(\boldsymbol{\pi}_{1:H}) := \underset{\tau \sim \boldsymbol{\pi}_{1:H}}{\mathbb{E}} \left[ \sum_{h=1}^H r(s_h, a_h, s_{h+1}) \right] $$

where $\tau$ refers to the sample trajectory $(s_1, a_1, s_2, a_2, \ldots, s_H)$ generated when following a policy sequence, i.e., $a_h \sim \pi_h(\cdot|s_h)$, $s_{h+1} \sim P(\cdot|s_h, a_h)$, and $s_1 \sim \mu$.

Let $R_{h:H}(\tau)$ denote the total reward of a trajectory $\tau$ starting from time $h$ until episode terminal time $H$ generated by following the policy sequence $\boldsymbol{\pi}_{h:H}$. We also define the state-value function of a state $s$ at step $h$ as $V_h^{\boldsymbol{\pi}}(s) := \underset{\tau \sim \boldsymbol{\pi}}{\mathbb{E}} [R_{h:H}(\tau) \,|\, s_h = s]$ and the action-value function as $Q_h^{\boldsymbol{\pi}}(s, a) := \underset{\tau \sim \boldsymbol{\pi}}{\mathbb{E}} [R_{h:H}(\tau) \,|\, s_h = s, a_h = a]$. The advantage function is $A_h^{\boldsymbol{\pi}}(s, a) := Q_h^{\boldsymbol{\pi}}(s, a) - V_h^{\boldsymbol{\pi}}(s)$. We also define $\mathbb{P}_h^{\boldsymbol{\pi}}(s \,|\, s_1) = \sum_{a \in A} \mathbb{P}_h^{\boldsymbol{\pi}}(s, a \,|\, s_1)$, where the term $\mathbb{P}_h^{\boldsymbol{\pi}}(s, a \,|\, s_1)$ is the probability of reaching $(s, a)$ at time step $h$ following $\boldsymbol{\pi}$ and starting from $s_1$.

**Constrained MDPs.** A constrained Markov decision process (CMDP) is an MDP augmented with constraints that restrict the set of allowable policies for that MDP. Specifically, we have $m$ cost functions, $C_1, \cdots, C_m$ (with each function $C_i : S \times A \times S \to \mathbb{R}$ mapping transition tuples to costs, similar to the reward function), and bounds $d_1, \cdots, d_m$. And similar to the value function for the reward objective, we define the expected total cost objective for each cost function $C_i$ (called cost value for the constraint) as

$$ J_{C_i}(\boldsymbol{\pi}_{1:H}) := \underset{\tau \sim \boldsymbol{\pi}_{1:H}}{\mathbb{E}} \left[ \sum_{h=1}^H C_i(s_h, a_h, s_{h+1}) \right]. $$

The goal then, in each episode, is to find a policy sequence $\boldsymbol{\pi}_{1:H}^\star$ such that

$$ J(\boldsymbol{\pi}_{1:H}^\star) := \max_{\boldsymbol{\pi}_{1:H} \in \Pi_C} J(\boldsymbol{\pi}_{1:H}), \ \ \text{where} \ \ \Pi_C := \{\boldsymbol{\pi}_{1:H} \in \Pi \ : \ J_{C_i}(\boldsymbol{\pi}_{1:H}) \leqslant d_i, \ \forall \, i \in [1:m]\} \quad (1) $$

is the set of feasible policies for a CMDP for some given class of policies $\Pi$. Lastly, analogous to $V_h^{\boldsymbol{\pi}}$, $Q_h^{\boldsymbol{\pi}}$, and $A_h^{\boldsymbol{\pi}}$, we can also define quantities for the cost functions $C_i(\cdot)$ by replacing, and denote them by $V_{C_i,h}^{\boldsymbol{\pi}}$, $Q_{C_i,h}^{\boldsymbol{\pi}}$, and $A_{C_i,h}^{\boldsymbol{\pi}}$. Proofs of theorems and statements, if not given, are available in Appendix A.

**Notation.** $[N]$ denotes $\{1, \ldots, N\}$ for some $N \in \mathbb{N}$. $\pi_h$ refers to the policy at time step $h$ within an episode. Denote $\boldsymbol{\pi}_{s:t} := (\pi_s, \pi_{s+1}, \ldots, \pi_t)$ for some $s \leqslant t$ with $s, t \in [H]$. We shall only write $\pi_{k,h}$ when it is necessary to differentiate policies from different episodes but at the same time $h$. It then naturally follows to define $\boldsymbol{\pi}_{k,s:t}$ to be the sequence $\boldsymbol{\pi}_{s:t}$ in episode $k$. We will denote $\boldsymbol{\pi}_k \equiv \boldsymbol{\pi}_{k,1:H}$, and where not needed drop the index for the episode so that $\boldsymbol{\pi} \equiv \boldsymbol{\pi}_k$.

## 3   Episodic Constrained Optimization of Policies (`e-COP`)

In this section, we propose a constrained policy optimization algorithm for episodic MDPs. Policy optimization algorithms for MDPs have proven remarkably effective given their ability to computationally scale up to high dimensional continuous state and action spaces [31–33]. Such algorithms have also been proposed for infinite-horizon constrained MDPs with discounted criterion [1] as well as the average criterion [4] but not for the finite horizon (or as it is often called, the episodic) setting.

We note that finite horizon is not simply a special case of infinite-horizon discounted setting since the reward/cost functions in the former can be time-varying while the latter only allows for time-invariant objectives. Furthermore, even with time-invariant objectives, the optimal policy is time-dependent, while for the latter setting there an optimal policy that is stationary exists.

**A Policy Difference Lemma for Episodic MDPs.** Most policy optimization RL algorithms are based on a value or policy difference technical lemma [23]. Unfortunately, the policy difference lemmas that have been derived previously for the infinite-horizon discounted [31] and average case [4] are not applicable here and hence, we derive a new policy difference lemma for the episodic setting.

**Lemma 3.1.** *For an episode of length $H$ and two policies, $\boldsymbol{\pi}$ and $\boldsymbol{\pi}'$, the difference in their performance assuming identical initial state distribution $\mu$ (i.e., $s_1 \sim \mu$) is given by*

$$J(\boldsymbol{\pi}) - J(\boldsymbol{\pi}') = \sum_{h=1}^{H} \mathop{\mathbb{E}}_{\substack{s_h, a_h \sim \mathbb{P}_h^{\boldsymbol{\pi}}(\cdot, \cdot \mid s) \\ s_1 \sim \mu}} \left[ A_h^{\boldsymbol{\pi}'}(s_h, a_h) \right]. \tag{2}$$

The proof can be found in Appendix A.1. A key difference to note between the above and similar results for infinite-horizon settings [31, 4] is that considering stationary policies (and hence corresponding occupation measures) is not enough for the episodic setting since, in general, such a policy may be far from optimal. This explains why Lemma 3.1 looks so different (e.g., see (2) in [31], and Lemma 3.1 in [4]). Indeed, the lemma above indicates that policy updates do not have to recurse backwards from the terminal time as dynamic programming algorithms do for episodic settings, which is somewhat surprising.

**A Constrained Policy Optimization Algorithm for Episodic MDPs.** Iterative policy optimization algorithms achieve state of the art performance [33, 36, 39] on RL problems. Most such algorithms maximize the advantage function based on a suitable policy difference lemma, solving an unconstrained RL problem. Some additionally ensure satisfaction of infinite horizon expectation constraints [3, 4]. However, given that our policy lemma for the episodic setting (Lemma 3.1) is significantly different, we need to re-design the algorithm based on it. A first attempt is presented as Algorithm 1, where each iteration corresponds to an update with a full horizon $H$ episode.

---

**Algorithm 1** **I**terative **P**olicy **O**ptimization for **C**onstrained **E**pisodic (IPOCE) RL

---

1: **Input:** Initial policy $\boldsymbol{\pi}_0$, number of episodes $K$, episode horizon $H$.
2: **for** $k = 1, 2, \ldots, K$ **do**
3:     Run $\boldsymbol{\pi}_{k-1}$ to collect trajectories $\tau$.
4:     Evaluate $A_h^{\boldsymbol{\pi}_{k-1}}$ and $A_{C_i,h}^{\boldsymbol{\pi}_{k-1}}$ for $h \in [H]$ from $\tau$.
5:     **for** $t = H, H-1, \ldots, 1$ **do**

6:       $\pi_{k,t}^{\star} = \underset{\pi_{k,t}}{\arg\min} \sum_{h=t}^{H} \mathop{\mathbb{E}}_{\substack{s \sim \rho_{\pi_{k,h}} \\ a \sim \pi_{k,h}}} \left[ -A_h^{\boldsymbol{\pi}_{k-1}}(s, a) \right] \quad \text{s.t.} \quad J_{C_i}(\boldsymbol{\pi}_{k-1}) + \sum_{h=t}^{H} \mathop{\mathbb{E}}_{\substack{s \sim \rho_{\pi_{k,h}} \\ a \sim \pi_{k,h}}} \left[ A_{C_i,h}^{\boldsymbol{\pi}_{k-1}}(s, a) \right] \leqslant d_i, \ \forall i \quad$ (3)

7:     **end for**
8:     Set $\boldsymbol{\pi}_k \leftarrow \left( \pi_{k,1}^{\star}, \pi_{k,2}^{\star}, \ldots, \pi_{k,H}^{\star} \right)$.
9: **end for**

---

The iterative constrained policy optimization algorithm introduced above uses the current iterate of the policy $\boldsymbol{\pi}_k$ to collect a trajectory $\tau$, and use them to evaluate $A_h^{\boldsymbol{\pi}_{k-1}}$ and $A_{C_i,h}^{\boldsymbol{\pi}_{k-1}}$ for $h \in [H]$. At the end of the episode, we solve $H$ optimization problems (one for each $h \in [H]$) that result in a new sequence of policies $\boldsymbol{\pi}$. As is natural in episodic problems, we do backward iteration in time, i.e., solve the problem in step (6) at $h = H$, and then go backwards towards $h = 1$.

Note that the expectation of advantage functions in equation (3) is with respect to the policy $\pi$ (the optimization variable) and its corresponding *time-dependent* state occupation distribution $\rho_{\pi_h}$. In the infinite-horizon settings, the expectation is with respect to the steady state stationary distribution, but that does not exist in the episodic setting.

**Using current policy for action selection.** Algorithm 1 represents an exact principled solution to the constrained episodic MDP, but the intractable optimization performed in (3) makes it impractical (as in the case of infinite horizon policy optimization algorithms [31, 3, 39]). We proceed to introduce a sequence of ideas that make the algorithm practical (e.g., by avoiding computationally expensive Hessian matrix inversion for use with trust-region methods [33, 14, 41]). However, getting rid of trust regions leads to large updates on policies, but PPO [33] and P3O [41] successfully overcome this problem by clipping the advantage function and adding a ReLU-like penalty term to the minimization objective. Motivated by this, we rewrite the optimization problem in (3) as follows by parameterizing the policy $\boldsymbol{\pi}_{k,t}$ in episode $k$ and time step $t$ by $\theta_{k,t}$:

$$\pi_{k,t} = \underset{\pi_{k,t}}{\arg\min} \sum_{h=t}^{H} \mathop{\mathbb{E}}_{\substack{s \sim \rho_{\pi_{k,h}} \\ a \sim \pi_{k-1,h}}} \left[ -\rho(\theta_h) A_h^{\boldsymbol{\pi}_{k-1}}(s, a) \right] + \sum_{i}^{m} \lambda_{t,i} \max \left\{ 0, \ \sum_{h=t}^{H} \mathop{\mathbb{E}}_{\substack{s \sim \rho_{\pi_{k,h}} \\ a \sim \pi_{k-1,h}}} \left[ \rho(\theta_h) A_{C_i,h}^{\boldsymbol{\pi}_{k-1}}(s, a) \right] + J_{C_i}(\boldsymbol{\pi}_{k-1}) - d_i \right\},$$
$$\tag{4}$$

where $\rho(\theta_h) = \frac{\pi_{\theta_{k,h}}}{\pi_{\theta_{k-1,h}}}$ is the importance sampling ratio, $\lambda_{t,i}$ is a penalty factor for constraint $C_i$, and $\pi_{k,\theta_h} \equiv \pi_{k,h} \equiv \theta_{k,h}$. Note that the ReLU-like penalty term above is different from the traditional first-order and second-order gradient approximations that are employed in trust-region methods [3, 39]. In essence, the penalty is applied when the agent breaches the associated constraint, while the objective remains consistent with standard policy optimization when all constraints are satisfied.

**Introducing quadratic damping penalty.** It has been noted in such iterative policy optimization algorithms that the behaviour of the learning agent when it nears the constraint threshold is quite volatile during training [3, 39, 42]. This is because the penalty term is active only when the constraints are violated which results in sharp behavior change for the agent. To alleviate this problem, we introduce an additional quadratic damping term to the objective above, which provides stable cost control to compliment the lagged Lagrangian multipliers. This has proved effective in physics-based control applications [16, 25, 20] resulting in improved convergence since the damping term provides stability, while keeping the solution set the same as for the original Problem (3) and the adapted Problem (4) (as we prove later).

For brevity, we denote the constraint term in Problem (4) as

$$\Psi_{C_i,t}(\boldsymbol{\pi}_{k-1}, \boldsymbol{\pi}_k) := \sum_{h=t}^{H} \mathbb{E}_{\substack{s \sim \rho_{\pi_{k,h}} \\ a \sim \pi_{k-1,h}}} \left[ \rho(\theta_h) A_{C_i,h}^{\boldsymbol{\pi}_{k-1}}(s,a) \right] + J_{C_i}(\boldsymbol{\pi}_{k-1}) - d_i.$$

Now introduce a slack variable $x_{t,i} \geqslant 0$ for each constraint to convert the inequality constraint $(\Psi_{C_i,t}(\cdot,\cdot) \leqslant 0)$ to equality by letting

$$w_{t,i}(\boldsymbol{\pi}_k) := \Psi_{C_i,t}(\boldsymbol{\pi}_{k-1}, \boldsymbol{\pi}_k) + x_{t,i} = 0.$$

With this notation, we restate Problem (4) as:

$$\pi_{k,t}^{\star} = \min_{\pi_{k,t}} \ \mathcal{L}_t(\boldsymbol{\pi}_k, \boldsymbol{\lambda}) := \sum_{h=t}^{H} \mathbb{E}_{\substack{s \sim \rho_{\pi_{k,h}} \\ a \sim \pi_{k-1,h}}} \left[ -\rho(\theta_h) A_h^{\boldsymbol{\pi}_{k-1}}(s,a) \right] + \sum_i^m \lambda_{t,i} \max\{0, \Psi_{C_i,t}(\boldsymbol{\pi}_{k-1}, \boldsymbol{\pi}_k)\}.$$

Now we introduce the quadratic damping term and the intermediate loss function then takes the form,

$$\mathcal{L}_t(\boldsymbol{\pi}_k, \boldsymbol{\lambda}, \boldsymbol{x}, \beta) := \sum_{h=t}^{H} \mathbb{E}_{\substack{s \sim \rho_{\pi_{k,h}} \\ a \sim \pi_{k-1,h}}} \left[ -\rho(\theta_h) A_h^{\boldsymbol{\pi}_{k-1}}(s,a) \right] + \sum_i^m \lambda_{t,i} w_{t,i}(\boldsymbol{\pi}_k) + \frac{\beta}{2} \sum_i^m w_{t,i}^2(\boldsymbol{\pi}_k) \tag{5}$$

$$\text{Then,} \qquad (\pi_{k,t}^{\star}, \boldsymbol{\lambda}_t^{\star}, \boldsymbol{x}_t^{\star}) = \max_{\boldsymbol{\lambda} \geqslant 0} \min_{\pi_{k,t}, \boldsymbol{x}} \mathcal{L}_t(\boldsymbol{\pi}_k, \boldsymbol{\lambda}, \boldsymbol{x}, \beta) ,$$

where $\beta$ is the damping factor, $\boldsymbol{\lambda}_t = (\lambda_{t,i})_{i=1}^m$, and $\boldsymbol{x}_t = (x_{t,i})_{i=1}^m$. We can then construct a primal-dual solution to the max-min optimization problem. The need for a slack variable $\boldsymbol{x}$ can be obviated by setting the partial derivative of $\mathcal{L}_t(\cdot)$ with respect to $\boldsymbol{x}$ equal to 0. This leads to a ReLU-like solution: $x_{t,i}^{\star} = \max\left(0, -\Psi_{C_i,t}(\boldsymbol{\pi}_{k-1}, \boldsymbol{\pi}_k) - \frac{\lambda_{t,i}}{\beta}\right)$. The intermediate problem then takes the form as below.

**Proposition 3.2.** *The inner optimization problem in* (5) *with respect to $\boldsymbol{x}$ is a convex quadratic program with non-negative constraints, which can be solved to yield the following intermediate problem:*

$$(\pi_{k,t}^{\star}, \boldsymbol{\lambda}_t^{\star}) = \max_{\boldsymbol{\lambda} \geqslant 0} \min_{\pi_{k,t}} \mathcal{L}_t(\boldsymbol{\pi}_k, \boldsymbol{\lambda}, \beta), \quad where$$

$$\mathcal{L}_t(\boldsymbol{\pi}_k, \boldsymbol{\lambda}, \beta) = \sum_{h=t}^{H} \mathbb{E}_{\substack{s \sim \rho_{\pi_{k,h}} \\ a \sim \pi_{k-1,h}}} \left[ -\rho(\theta_h) A_h^{\boldsymbol{\pi}_{k-1}}(s,a) \right] + \frac{\beta}{2} \sum_i^m \left( \max\left\{ 0, \Psi_{C_i,t}(\boldsymbol{\pi}_{k-1}, \boldsymbol{\pi}_k) + \frac{\lambda_{t,i}}{\beta} \right\}^2 - \frac{\lambda_{t,i}^2}{\beta^2} \right). \tag{6}$$

The proof can be found in Appendix A.1. One can see that the cost penalty is active when $\Psi_{C_i,t}(\boldsymbol{\pi}_{k-1}, \boldsymbol{\pi}_k) \geqslant -\frac{\lambda_{t,i}}{\beta}$ rather than when $\Psi_{C_i,t}(\boldsymbol{\pi}_{k-1}, \boldsymbol{\pi}_k) \geqslant 0$. This allows the agent to act in a constrained manner even before the constraint is violated. Further, as we show next, the introduction of the damping factor and the RELU-like penalty does not change the solution of the problem (under some suitable assumptions):

**Theorem 3.3.** *Let $\pi^{(3)^{\star}}$ be a solution to Problem* (3)*, and let $\left(\pi^{(6)^{\star}}, \boldsymbol{\lambda}^{(6)^{\star}}\right)$ be a solution to Problem* (6)*. Then, for sufficiently large $\beta > \bar{\beta}$ and $\lambda_{t,i} > \bar{\lambda} \ \forall \ i$, $\pi^{(3)^{\star}}$ is a solution to Problem* (6)*, and $\pi^{(6)^{\star}}$ is a solution to Problem* (3)*.*

We refer the reader to Appendix A.1 for the proof. This theorem implies that we can search for the optimal feasible policies of the CMDP Problem (1) by iteratively solving Problem (6). Next, we make some further modifications to Problem (6) that give us our final tractable algorithm.

**Removing Lagrange multiplier dependency.** Problem (6) requires a primal-dual algorithm that will iteratively solve over the policies and the dual variable $\lambda$. But from the Lagrangian, we can actually take a derivative with respect to $\lambda$, and then solve for it, which yields the following update rule for it:

$$\lambda_{t,i}^{(k)} = \max\left(0, \lambda_{t,i}^{(k-1)} + \beta\Psi_{C_i,t}(\boldsymbol{\pi}_{k-1}, \boldsymbol{\pi}_{k-1})\right). \tag{7}$$

This update rule simplifies the optimization problem and updates the Lagrange multipliers in the *kth* episode based on the constraint violation in the $(k-1)th$ episode.

**Clipping the advantage functions.** Solving the optimization problem presented in equation (6) is intractable since we do not know $\rho_\pi$ beforehand. Hence, we replace $\rho_\pi$ by the empirical distribution observed with the policy of the previous episode, $\boldsymbol{\pi}_{k-1}$, i.e., $\rho_{\pi_{k,h}} \approx \rho_{\pi_{k-1,h}} \ \forall \ h$. Similar to [33] for PPO, we also use *clipped* surrogate objective functions for both the reward and cost advantage functions. Thus, the final problem combining equation (4) and equation (6) can be constructed as follows.

If we let

$$\mathcal{L}_t(\theta) = \sum_{h=t}^{H} \mathbb{E}_{\substack{s\sim\rho_{\pi_{k-1,h}} \\ a\sim\pi_{k-1,h}}} \left[ -\min\left\{\rho(\theta_h)A_h^{\boldsymbol{\pi}_{k-1}}(s,a), \mathrm{clip}(\rho(\theta_h), 1-\epsilon, 1+\epsilon)A_h^{\boldsymbol{\pi}_{k-1}}(s,a)\right\}\right] \quad \text{and,}$$

$$\mathcal{L}_{C_i,t}(\theta) = \sum_{h=t}^{H} \mathbb{E}_{\substack{s\sim\rho_{\pi_{k-1,h}} \\ a\sim\pi_{k-1,h}}} \left[ -\min\left\{\rho(\theta_h)A_{C_i,h}^{\boldsymbol{\pi}_{k-1}}(s,a), \mathrm{clip}(\rho(\theta_h), 1-\epsilon, 1+\epsilon)A_{C_i,h}^{\boldsymbol{\pi}_{k-1}}(s,a)\right\}\right]$$

then, the final loss function $\widetilde{\mathcal{L}}_t(\cdot)$ of the final problem is:

$$\begin{aligned}
\pi_{k,t}^{\star} = \underset{\pi_{k,t}}{\arg\min} \, \widetilde{\mathcal{L}}_t(\pi_\theta, \boldsymbol{\lambda}, \beta) &:= \underset{\pi_{k,t}}{\arg\min} \ \mathcal{L}_t(\theta) + \sum_i^m \lambda_{t,i}\max\left\{0, \mathcal{L}_{C_i,t}(\theta) + \left(J_{C_i}(\boldsymbol{\pi}_{k-1}) - d_i\right)\right\} \\
&+ \frac{\beta}{2}\sum_i^m \left(\max\left\{0, \mathcal{L}_{C_i,t}(\theta) + \left(J_{C_i}(\boldsymbol{\pi}_{k-1}) - d_i\right) + \frac{\lambda_{t,i}}{\beta}\right\}^2 - \frac{\lambda_{t,i}^2}{\beta^2}\right)
\end{aligned} \tag{8}$$

Usually for experiments, Gaussian policies with means and variances predicted from neural networks are used [31, 3, 33, 39]. We employ the same approach and since we work in the finite horizon setting, the reward and constraint advantage functions can easily be calculated from any trajectory $\tau \sim \boldsymbol{\pi}$. The surrogate problem in equation (8) then includes the pessimistic bounds on Problem (6), which is unclipped.

**Adaptive parameter selection.** The value of $\beta$ is required to be larger than the unknown $\bar{\beta}$ according to Theorem 3.3, but we also know that too large a $\beta$ decays the performance (as seen in harmonic oscillator kinetic energy formulations [25, 20, 40]). To manage this tradeoff, we provide an instance-dependent adaptive way to adjust the damping factor as a hyperparameter. In each episode $k$, we update the damping parameter whenever a secondary constraint cost value denoted by $\mathcal{C}(\boldsymbol{\pi}_k)$ is larger than some threshold $c_k$. Using Proposition 3.2, we provide the following definitions.

$$\mathcal{C}(\boldsymbol{\pi}_k) := \sum_{t=1}^{H}\sum_i^m \max\left\{J_{C_i}(\boldsymbol{\pi}_k) - d_i, -\frac{\lambda_{t,i}^{(k)}}{\beta}\right\} \quad \text{and} \quad c_k := \frac{\sqrt{m}}{\beta}\cdot\max_{t\in[H]}\left\|\boldsymbol{\lambda}_t^{(k)}\right\|_\infty$$

---

**Algorithm 2** **E**pisodic **C**onstrained **O**ptimization of **P**olicies (`e-COP`)
___________________________________________________________________________

1: **Input:** Initial policy $\theta_0 := \boldsymbol{\pi}_0 := \boldsymbol{\pi}_{\theta_0}$, critic networks $V^{\phi_0}$ and $V_{C_i}^{\psi_0} \ \forall \ i$, penalty factor $\beta$, number of episodes $K$, episode horizon $H$, learning rate $\alpha$, penalty update factor $p$.
2: **for** $k = 1, 2, \ldots, K$ **do**
3:     Collect a set of trajectories $\mathcal{D}_{k-1}$ with policy $\boldsymbol{\pi}_{k-1}$ and update the critic network.
4:     Get updated $\lambda^{(k)}$ using equation (7).
5:     **for** $t = H, H-1, \ldots, 1$ **do**
6:         Update the policy $\theta_{k,t} \leftarrow \theta_{k,t+1} - \alpha\nabla_\theta\widetilde{\mathcal{L}}_t(\theta, \lambda^{(k)}, \beta)$ using equation (8).
7:     **end for**
8:     **if** $\mathcal{C}(\theta_k) \geqslant c_k$ **then**
9:         $\beta = \min(\beta_{\max}, p\beta)$
10:     **end if**
11: **end for**
___________________________________________________________________________

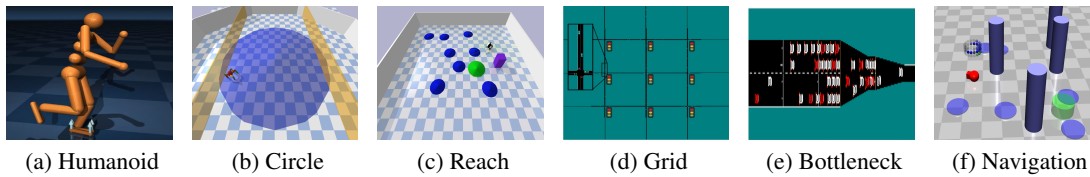

| (a) Humanoid | (b) Circle | (c) Reach | (d) Grid | (e) Bottleneck | (f) Navigation |

Figure 1: The Humanoid, Circle, Reach, Grid, Bottleneck, and Navigation tasks. See Appendix A.2.1 for details.

Hence, we increase $\beta$ by a constant factor $\rho > 1$ after every episode if $C(\pi_k) \geqslant c_k$ until a stopping condition is fulfilled, typically a constant $\beta_{\max}$. This leads to constraint-satisfying iterations that are more stable, and we show that it enables a fixed $\beta$ to generalize well across various tasks. The initial $\beta$ can simply be selected by a quantified line-search to obtain a feasible $\beta > \bar{\beta}$ [3, 4].

We note that the final loss function in equation (8) is differentiable almost everywhere, so we could easily solve it via any first-order optimizer [24]. The final practical algorithm, e-COP is given in Algorithm 2.

## 4 Experiments

We conducted extensive experimental evaluation on the relative empirical performance of the e-COP algorithm to arrive at the following conclusions: (i) The e-COP algorithm performs better or nearly as well as all baseline algorithms for infinite-horizon discounted safe RL tasks in maximizing episodic return while satisfying given constraints. (ii) It is more robust to stochastic and complex environments [30], even where previous methods struggle. (iii) It has stable behavior and more accurate cost control as compared to other baselines near the constraint threshold due to the damping term.

**Environments.** For a comprehensive empirical evaluation, we selected eight scenarios from well-known safe RL benchmark environments - Safe MuJoCo [43] and Safety Gym [30], as well as MuJoCo environments. These include: `Humanoid`, `PointCircle`, `AntCircle`, `PointReach`, `AntReach`, `Grid`, `Bottleneck`, and `Navigation`. See Figure 1 for an overview of the tasks and scenarios. Note that `Navigation` is a multi-constraint task and for the `Reach` environment, we set the reward as a function of the Euclidean distance between agent's position and goal position. In addition, we make it impossible for the agent to reach the goal before the end of the episode. For more information see Appendix A.2.1.

**Baselines.** We compare our e-COP algorithm with the following baseline algorithms: CPO [3], PCPO [39], FOCOPS [43], PPO with Lagrangian relaxation [33, 35], and penalty-based P3O [41]. Since the above state-of-the-art baseline algorithms are already well understood to outperform other algorithms such as PDO [13], IPO [26], and CPPO-PID [35] in prior benchmarking studies, we do not compare against them. Moreover, since PPO does not originally incorporate constraints, for fair comparison, we introduce constraints using a Lagrangian relaxation (called PPO-L). In addition, for each algorithm, we report its performance with the discount factor that achieves the best performance. See Appendix A.3.1 for more details.

**Evaluation Details and Protocol.** For the Circle task, we use a a point-mass with $S \subseteq \mathbb{R}^9, A \subseteq \mathbb{R}^2$ and for the Reach task, an ant robot with $S \subseteq \mathbb{R}^{16}, A \subseteq \mathbb{R}^8$. The Grid task has $S \subseteq \mathbb{R}^{56}, A \subseteq \mathbb{R}^4$. We use two hidden layer neural networks to represent Gaussian policies for the tasks. For Circle and Reach, size is (32,32) for both layers, and for Grid and Navigation the layer sizes are (16,16) and (25,25). We set the step size $\delta$ to $10^{-4}$, and for each task, we conduct 5 independent runs of $K = 500$ episodes each of horizon $H = 200$. Since there are two objectives (rewards in the objective and costs in the constraints), we show the plots which maximize the reward objective while satisfying the cost constraint.

### 4.1 Performance Analysis

Table 1 lists the numerical performance of all tested algorithms in seven single constraint scenarios, and one multiple constraint scenario. We find that overall, the e-COP algorithm in most cases outperforms (green) all other baseline algorithms in finding the optimal policy while satisfying the constraints, and in other cases comes a close second (light green).

From Figure 2, we can see how the e-COP algorithm is able to improve the reward objective over the baselines while having approximate constraint satisfaction. We also see that updates of e-COP are faster and smoother than other baselines due to the added damping penalty, which ensures smoother

| Task | | e-COP | FOCOPS [43] | PPO-L [30] | PCPO [39] | P3O [41] | CPO [3] | APPO [15] | IPO [26] |
|---|---|---|---|---|---|---|---|---|---|
| Humanoid | R | 1652.5 ± 13.4 | **1734.1 ± 27.4** | 1431.2 ± 25.2 | 1602.3 ± 10.1 | 1669.4 ± 13.7 | 1465.1 ± 55.3 | 1488.2 ± 29.3 | 1578.6 ± 25.2 |
| | C (20.0) | 17.3 ± 0.3 | 19.7 ± 0.6 | 18.8 ± 1.5 | 16.3 ± 1.4 | 20.1 ± 3.3 | 18.5 ± 2.9 | 20.0 ± 1.3 | 19.1 ± 2.5 |
| PointCircle | R | **110.5 ± 9.3** | 81.6 ± 8.4 | 57.2 ± 9.2 | 68.2 ± 9.1 | 89.1 ± 7.1 | 65.3 ± 5.3 | 91.2 ± 9.6 | 68.7 ± 15.2 |
| | C (10.0) | 9.8 ± 0.9 | 10.0 ± 0.4 | 9.8 ± 0.5 | 9.9 ± 0.4 | 9.9 ± 0.3 | 9.5 ± 0.9 | 10.2 ± 0.6 | 9.3 ± 0.5 |
| AntCircle | R | **198.6 ± 7.4** | 161.9 ± 22.2 | 134.4 ± 10.3 | 168.3 ± 13.3 | 182.6 ± 18.7 | 127.1 ± 12.1 | 155.5 ± 19.4 | 149.3 ± 33.6 |
| | C (10.0) | 9.8 ± 0.6 | 9.9 ± 0.5 | 9.6 ± 1.6 | 9.5 ± 0.6 | 9.8 ± 0.2 | 10.1 ± 0.7 | 10.0 ± 0.5 | 9.5 ± 1.0 |
| PointReach | R | **81.5 ± 10.2** | 65.1 ± 9.6 | 46.1 ± 14.8 | 73.2 ± 7.4 | 76.3 ± 6.4 | 89.2 ± 8.1 | 74.3 ± 6.7 | 49.1 ± 10.6 |
| | C (25.0) | 24.5 ± 6.1 | 24.8 ± 7.6 | 25.1 ± 6.1 | 24.9 ± 5.6 | 26.3 ± 6.9 | 33.3 ± 10.7 | 26.3 ± 8.1 | 24.7 ± 11.3 |
| AntReach | R | 70.8 ± 14.6 | 48.3 ± 5.6 | 54.2 ± 9.5 | 39.4 ± 5.3 | **73.6 ± 5.1** | 102.3 ± 7.1 | 61.5 ± 10.4 | 45.2 ± 13.3 |
| | C (25.0) | 24.2 ± 8.4 | 25.1 ± 11.9 | 21.9 ± 10.7 | 27.9 ± 12.2 | 24.8 ± 7.3 | 35.1 ± 10.9 | 24.5 ± 6.4 | 24.9 ± 9.2 |
| Grid | R | 258.1 ± 33.1 | 215.4 ± 45.6 | **276.3 ± 57.9** | 226.5 ± 29.2 | 201.5 ± 39.2 | 178.1 ± 23.8 | 184.4 ± 21.5 | 229.4 ± 32.8 |
| | C (75.0) | 71.3 ± 26.9 | 76.6 ± 29.8 | 71.8 ± 25.1 | 72.6 ± 16.5 | 79.3 ± 19.3 | 69.3 ± 19.8 | 79.5 ± 35.8 | 74.2 ± 24.6 |
| Bottleneck | R | **345.1 ± 52.6** | 251.3 ± 59.1 | 298.3 ± 71.2 | 264.2 ± 43.8 | 291.1 ± 26.7 | 388.1 ± 36.6 | 220.1 ± 30.1 | 279.3 ± 43.8 |
| | C (50.0) | 49.7 ± 15.1 | 46.6 ± 19.8 | 41.4 ± 17.6 | 49.8 ± 10.5 | 45.3 ± 8.2 | 54.3 ± 13.5 | 47.4 ± 12.3 | 48.2 ± 14.6 |
| Navigation | R | **217.6 ± 11.5** | | 175.1 ± 3.7 | | 153.5 ± 25.2 | | 135.7 ± 19.2 | 164.1 ± 12.8 |
| | C1 (10.0) | 9.6 ± 1.5 | n/a | 9.9 ± 1.9 | n/a | 9.9 ± 1.7 | n/a | 9.9 ± 2.1 | 10.0 ± 0.5 |
| | C2 (25.0) | 23.7 ± 4.1 | | 22.3 ± 2.1 | | 24.5 ± 4.1 | | 23.9 ± 3.8 | 24.6 ± 3.1 |

Table 1: Mean performance with normal 95% confidence interval over 5 independent runs on some tasks.

**Episodic Rewards:**

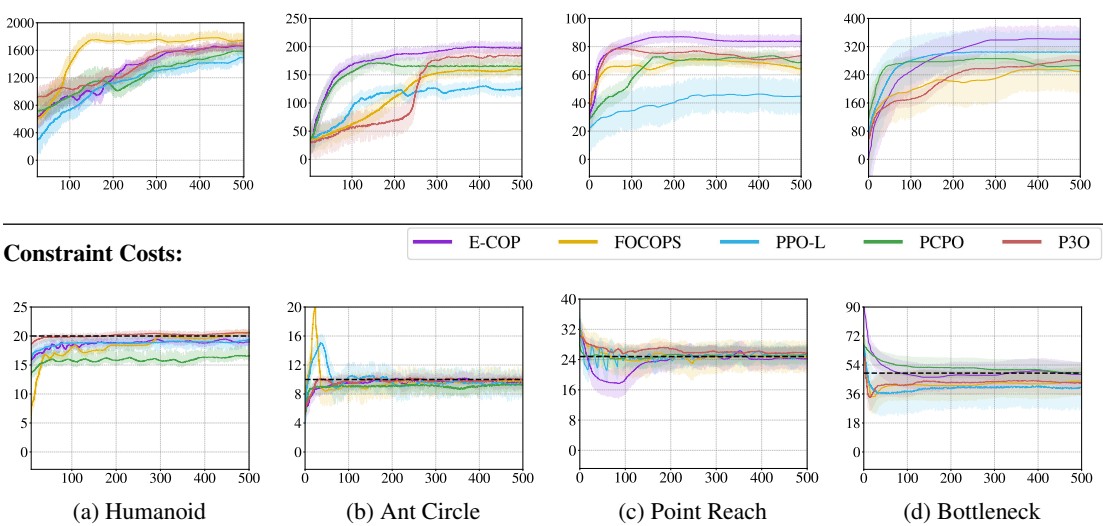

**Constraint Costs:**

(a) Humanoid  (b) Ant Circle  (c) Point Reach  (d) Bottleneck

Figure 2: The cumulative episodic reward and constraint cost function values vs episode learning curves for some algorithm-task pairs. Solid lines in each figure are the means, while the shaded area represents 1 standard deviation, all over 5 runs. The dashed line in constraint plots is the constraint threshold.

convergence with only a few constraint-violating behaviors during training. In particular, e-COP is the *only* algorithm that best learns almost-constraint-satisfying maximum reward policies across *all* tasks: in simple Humanoid and Circle environments, e-COP is able to almost exactly track the cost constraint values to within the given threshold. However, for the high dimensional Grid environment we have more constraint violations due to complexity of the policy behavior, leading to higher variance in episodic rewards as compared to other environments. Regardless, overall in these environments, e-COP still outperforms *all* other baselines with the least episodic constraint violation. For the multiple constraint Navigation environment, see Figure 3.

## 4.2 Secondary Evaluation

In this section, we take a deeper dive into the empirical performance of e-COP. We discuss its dependence on various factors, and try to verify its merits.

**Generalizability.** From the discussion above, it's clear that e-COP demonstrates accurate safety satisfaction across tasks of varying difficulty levels. From Figure 4, we further see that e-COP satisfies the

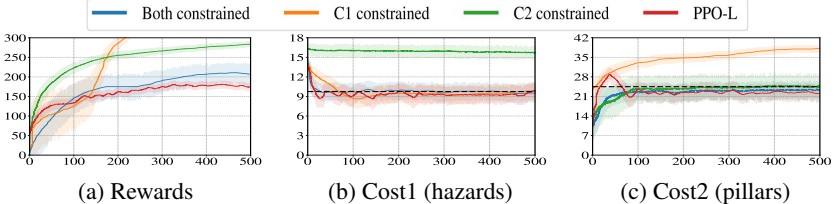

Figure 3: `Navigation` environment with multiple constraints: Episodic Rewards (left), Cost1 (center, for hazards) and Cost2 (right, for pillars) of `e-COP`. The dashed line in the cost plots is the cost threshold (10 for Cost1 and 25 for Cost2). C1/C2 constrained means only taking Cost1/Cost2 into the `e-COP` loss function and ignoring the other one.

constraints in all cases and precisely converges to the specified cost limit. Furthermore, the fluctuation observed in the baseline Lagrangian-based algorithms is shown not to be tied to a specific cost limit.

We also conducted a set of experiments wherein we study how `e-COP` effectively adapts to different cost thresholds. For this, we use the hyperparameters of a pre-trained `e-COP` agent, which is trained with a particular cost threshold in an environment, for learning on different cost thresholds within the same environment. Figure 6 in Appendix A.3.4 illustrates the training curves of these pre-trained agents, and we see that while `e-COP` can generalize well across different cost thresholds, other baseline algorithms may require further tuning to accommodate different constraint thresholds.

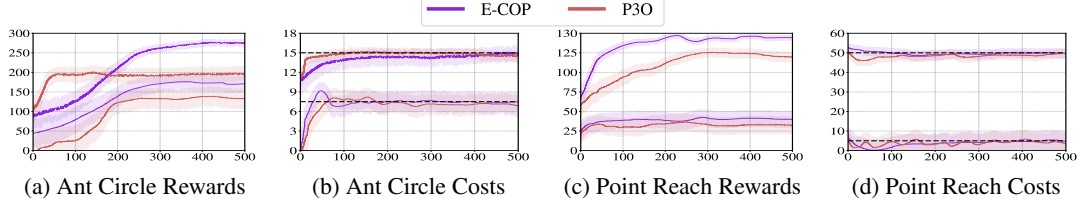

| (a) Ant Circle Rewards | (b) Ant Circle Costs | (c) Point Reach Rewards | (d) Point Reach Costs |

Figure 4: Cumulative episodic rewards and costs of baselines in two environments with two constraint thresholds.

| Task | | e-COP | | | | P3O [41] |
|------|---|-------|---|---|---|----------|
| | | $\beta = 5$, fixed | $\beta = 5$, adaptive | $\beta = 10$, fixed | $\beta = 10$, adaptive | |
| PointCircle | R | $150.5 \pm 11.1$ | $\mathbf{168.6 \pm 14.3}$ | $145.2 \pm 12.2$ | $165.3 \pm 11.4$ | $162.4 \pm 14.7$ |
| | C (20.0) | $17.3 \pm 1.3$ | $19.7 \pm 0.6$ | $18.8 \pm 1.5$ | $18.3 \pm 1.4$ | $19.1 \pm 3.3$ |
| AntReach | R | $48.2 \pm 3.5$ | $58.6 \pm 5.1$ | $53.2 \pm 5.3$ | $\mathbf{65.2 \pm 8.1}$ | $61.1 \pm 5.6$ |
| | C (20.0) | $19.8 \pm 5.9$ | $20.0 \pm 4.4$ | $20.6 \pm 4.5$ | $19.2 \pm 6.2$ | $18.9 \pm 7.3$ |

Table 2: Performance of `e-COP` for different $\beta$ settings on two tasks. Values are given with normal 95% confidence interval over 5 independent runs.

**Sensitivity.** The effectiveness and performance of `e-COP` would not be justified if it was not robust to the damping hyperparameter $\beta$, which varies across tasks depending on the values of $\mathcal{C}(\cdot)$ and $c_h$. Since this damping penalty enables `e-COP` to have stable continuous cost control, we update it adaptively as described in Algorithm 2. As seen in Table 2, damping penalty indeed stabilizes the training process and helps in converging to an optimal safe policy.

## 5 Conclusion

In this paper, we have introduced an easy to implement, scalable policy optimization algorithm for episodic RL problems with constraints due to safety or other considerations. It is based on a policy difference lemma for the episodic setting, which surprisingly has quite a different form than the ones for infinite-horizon discounted or average settings. This provides the theoretical foundation for the algorithm, which is designed by incorporating several time-tested, practical as well as novel ideas. Policy optimization algorithms for Constrained MDPs tend to be numerical unstable and non-scalable due to the need for inverting the Fisher information matrix. We sidestep both of these issues by introducing a quadratic damping penalty term that works remarkably well. The algorithm development is well supported by theory, as well as with extensive empirical analysis on a range of Safety Gym and Safe MuJoco benchmark environments against a suite of baseline algorithms adapted from their non-episodic roots.

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

# A  Appendix

## A.1  Proofs

**Lemma A.1.** *For an episode of length $H$ and two policies, $\boldsymbol{\pi}$ and $\boldsymbol{\pi}'$, the difference in their performance assuming identical initial state distribution $\mu$ (i.e., $s_1 \sim \mu$) is given by*

$$J(\boldsymbol{\pi}) - J(\boldsymbol{\pi}') = \sum_{h=1}^{H} \underset{\substack{s_h,a_h \sim \mathbb{P}_h^{\boldsymbol{\pi}}(\cdot,\cdot \,|\, s) \\ s_1 \sim \mu}}{\mathbb{E}} \big[ A_h^{\boldsymbol{\pi}'}(s_h, a_h) \big]. \tag{2}$$

*Proof.* Let us consider $s_1 \sim \mu$ and categorize the value function difference between the two policies. Also define $\mathbb{P}_h^{\boldsymbol{\pi}}(s \,|\, s_1) = \sum_{a \in A} \mathbb{P}_h^{\boldsymbol{\pi}}(s, a \,|\, s_1)$, where the term $\mathbb{P}_h^{\boldsymbol{\pi}}(s, a \,|\, s_1)$ is the probability of reaching $(s, a)$ at time step $h$ following $\boldsymbol{\pi}$ and starting from $s_1$.

$$
\begin{aligned}
V_1^{\boldsymbol{\pi}}(s_1) - V_1^{\boldsymbol{\pi}'}(s_1) &= \underset{a_1,s_2}{\mathbb{E}} \big[ r(s_1,a_1) + V_2^{\boldsymbol{\pi}}(s_2)|s_1 \big] + \underset{s_2}{\mathbb{E}} \big[ V_2^{\boldsymbol{\pi}'}(s_2) - V_2^{\boldsymbol{\pi}'}(s_2)|s_1 \big] - V_1^{\boldsymbol{\pi}'}(s_1) \\
&= \underset{s_2}{\mathbb{E}} \big[ V_2^{\boldsymbol{\pi}}(s_2) - V_2^{\boldsymbol{\pi}'}(s_2)|s_1 \big] + \underset{a_1,s_2}{\mathbb{E}} \big[ r(s_1,a_1) + V_2^{\boldsymbol{\pi}'}(s_2) - V_1^{\boldsymbol{\pi}'}(s_1)|s_1 \big] \\
&= \underset{s_2}{\mathbb{E}} \big[ V_2^{\boldsymbol{\pi}}(s_2) - V_2^{\boldsymbol{\pi}'}(s_2)|s_1 \big] + \underset{a_1}{\mathbb{E}} \big[ Q_1^{\boldsymbol{\pi}'}(s_1,a_1) - V_1^{\boldsymbol{\pi}'}(s_1)|s_1 \big] \\
&= \underset{s_2}{\mathbb{E}} \big[ V_2^{\boldsymbol{\pi}}(s_2) - V_2^{\boldsymbol{\pi}'}(s_2)|s_1 \big] + \underset{a_1}{\mathbb{E}} \big[ A_1^{\boldsymbol{\pi}'}(s_1,a_1)|s_1 \big],
\end{aligned}
$$

where $a_1 \sim \boldsymbol{\pi}_1(\cdot|s_1)$, $s_2 \sim P(\cdot|s_1, \boldsymbol{\pi}_1(s_1))$ and $s_1 \sim \mu$, the initial state distribution.

Now recursively apply the same procedure to the term $V_h^{\boldsymbol{\pi}}(s_h) - V_h^{\boldsymbol{\pi}'}(s_h) \ \forall \, h \in \{2, \ldots, H\}$ to obtain the following:

$$V_1^{\boldsymbol{\pi}}(s) - V_1^{\boldsymbol{\pi}'}(s) = \sum_{h=1}^{H} \underset{s_h,a_h \sim \mathbb{P}_h^{\boldsymbol{\pi}}(\cdot,\cdot \,|\, s)}{\mathbb{E}} \big[ A_h^{\boldsymbol{\pi}'}(s_h, a_h)|s \big]$$

Now we know that $J(\boldsymbol{\pi}) = \underset{s \sim \mu}{\mathbb{E}} [V_1^{\boldsymbol{\pi}}(s)]$, this means that we combine this with the above to obtain the final result. $\square$

**Proposition A.2.** *The inner optimization problem in (5) with respect to $\boldsymbol{x}$ is a convex quadratic program with non-negative constraints, which can be solved to yield the following intermediate problem:*

$$(\boldsymbol{\pi}_{k,t}^{\star}, \boldsymbol{\lambda}_t^{\star}) = \max_{\boldsymbol{\lambda} \geqslant 0} \min_{\boldsymbol{\pi}_{k,t}} \mathcal{L}_t(\boldsymbol{\pi}_k, \boldsymbol{\lambda}, \beta), \quad \text{where}$$

$$\mathcal{L}_t(\boldsymbol{\pi}_k, \boldsymbol{\lambda}, \beta) = \sum_{h=t}^{H} \underset{\substack{s \sim \rho_{\pi_{k,h}} \\ a \sim \pi_{k-1,h}}}{\mathbb{E}} \big[ -\rho(\theta_h) A_h^{\boldsymbol{\pi}_{k-1}}(s, a) \big] + \frac{\beta}{2} \sum_i^{m} \bigg( \max \Big\{ 0, \Psi_{C_i,t}(\boldsymbol{\pi}_{k-1}, \boldsymbol{\pi}_k) + \frac{\lambda_{t,i}}{\beta} \Big\}^2 - \frac{\lambda_{t,i}^2}{\beta^2} \bigg). \tag{6}$$

*Proof.* As in Equation (4), we have the following equivalent problem:

$$\pi_{k,t}^{\star} = \arg\min_{\pi_{k,t}} \sum_{h=t}^{H} \underset{\substack{s \sim \rho_{\pi_{k,h}} \\ a \sim \pi_{k-1,h}}}{\mathbb{E}} \big[ -\rho(\theta_h) A_h^{\boldsymbol{\pi}_{k-1}}(s, a) \big] + \sum_i^{m} \lambda_{t,i} \max \Big\{ 0, \sum_{h=t}^{H} \underset{\substack{s \sim \rho_{\pi_{k,h}} \\ a \sim \pi_{k-1,h}}}{\mathbb{E}} \big[ \rho(\theta_h) A_{C_i,h}^{\boldsymbol{\pi}_{k-1}}(s, a) \big] + (J_{C_i}(\boldsymbol{\pi}_{k-1}) - d_i) \Big\}.$$

Letting $\Psi_{C_i,t}(\boldsymbol{\pi}_{k-1}, \boldsymbol{\pi}_k) := \sum_{h=t}^{H} \underset{\substack{s \sim \rho_{\pi_{k,h}} \\ a \sim \pi_{k-1,h}}}{\mathbb{E}} \big[ \rho(\theta_h) A_{C_i,h}^{\boldsymbol{\pi}_{k-1}}(s, a) \big] + (J_{C_i}(\boldsymbol{\pi}_{k-1}) - d_i)$, and introducing slack variables $x_{t,i} \geqslant 0$ and defining $w_{t,i}(\boldsymbol{\pi}_k) := \Psi_{C_i,t}(\boldsymbol{\pi}_{k-1}, \boldsymbol{\pi}_k) + x_{t,i} = 0$, we get the quadratic damped problem same as Equation (5) below.

$$
\begin{aligned}
(\pi_{k,t}^{\star}, \boldsymbol{\lambda}_t^{\star}, \boldsymbol{x}_t^{\star}) &= \max_{\boldsymbol{\lambda} \geqslant 0} \min_{\boldsymbol{\pi}_{k,t}, \boldsymbol{x}} \mathcal{L}_t(\boldsymbol{\pi}_k, \boldsymbol{\lambda}, \boldsymbol{x}, \beta) \\
&= \max_{\boldsymbol{\lambda} \geqslant 0} \min_{\boldsymbol{\pi}_{k,t}, \boldsymbol{x}} \sum_{h=t}^{H} \underset{\substack{s \sim \rho_{\pi_{k,h}} \\ a \sim \pi_{k-1,h}}}{\mathbb{E}} \big[ -\rho(\theta_h) A_h^{\boldsymbol{\pi}_{k-1}}(s, a) \big] + \sum_i^{m} \lambda_{t,i} w_{t,i}(\boldsymbol{\pi}_k) + \frac{\beta}{2} \sum_i^{m} w_{t,i}^2(\boldsymbol{\pi}_k)
\end{aligned}
\tag{9}
$$

Like the Lagrangian method, we can alternately update $\pi, \boldsymbol{\lambda}$, and $\boldsymbol{x}$ to find the optimal triplet. Consider updating $\pi$ and $\boldsymbol{x}$ by minimizing $\mathcal{L}_t(\pi, \boldsymbol{\lambda}, \boldsymbol{x}, \beta)$ at any iteration:

$$(\pi_{k,t}^{\star}, \boldsymbol{x}_t^{\star}) = \arg\min_{\pi_{k,t}} \min_{x_i>0} \sum_{h=t}^{H} \mathbb{E}_{\substack{s\sim\rho_{\pi_{k,h}} \\ a\sim\pi_{k-1,h}}} \left[ -\rho(\theta_h) A_h^{\pi_{k-1}}(s,a) \right] + \sum_{i=1}^{m} \lambda_{t,i}\left(\Psi_{C_i,t}(\boldsymbol{\pi}_{k-1}, \boldsymbol{\pi}_k) + x_{t,i}\right) + \frac{\beta}{2}\sum_{i=1}^{m}\left(\Psi_{C_i,t}(\boldsymbol{\pi}_{k-1}, \boldsymbol{\pi}_k) + x_{t,i}\right)^2$$

The inner optimization problem with respect to $\boldsymbol{x}$ is a convex quadratic programming problem with non-negative constraints,

$$\boldsymbol{x}_t^{\star} = \arg\min_{x_i>0} \sum_{i=1}^{m} \lambda_{t,i}\left(\Psi_{C_i,t}(\boldsymbol{\pi}_{k-1}, \boldsymbol{\pi}_k) + x_{t,i}\right) + \frac{\beta}{2}\sum_{i=1}^{m}\left(\Psi_{C_i,t}(\boldsymbol{\pi}_{k-1}, \boldsymbol{\pi}_k) + x_{t,i}\right)^2$$

The optimal solution is $x_{t,i}^{\star} = \max\left\{0, -\frac{\lambda_{t,i}}{\beta} - \Psi_{C_i,t}(\boldsymbol{\pi}_{k-1}, \boldsymbol{\pi}_k)\right\}$. Then,

$$\begin{aligned}
w_{t,i}(\boldsymbol{\pi}_k) = \Psi_{C_i,t}(\boldsymbol{\pi}_{k-1}, \boldsymbol{\pi}_k) + x_{t,i}^{\star} &= \Psi_{C_i,t}(\boldsymbol{\pi}_{k-1}, \boldsymbol{\pi}_k) + \max\left\{0, -\frac{\lambda_{t,i}}{\beta} - \Psi_{C_i,t}(\boldsymbol{\pi}_{k-1}, \boldsymbol{\pi}_k)\right\} \\
&= \frac{\lambda_{t,i}}{\beta} + \Psi_{C_i,t}(\boldsymbol{\pi}_{k-1}, \boldsymbol{\pi}_k) + \max\left\{0, -\frac{\lambda_{t,i}}{\beta} - \Psi_{C_i,t}(\boldsymbol{\pi}_{k-1}, \boldsymbol{\pi}_k)\right\} - \frac{\lambda_{t,i}}{\beta} \\
&= \max\left\{0, \frac{\lambda_{t,i}}{\beta} + \Psi_{C_i,t}(\boldsymbol{\pi}_{k-1}, \boldsymbol{\pi}_k)\right\} - \frac{\lambda_{t,i}}{\beta}
\end{aligned}$$

Substituting back into Equation (9), we get

$$\begin{aligned}
\mathcal{L}_t(\boldsymbol{\pi}_k, \boldsymbol{\lambda}, \boldsymbol{x}, \beta) &= \sum_{h=t}^{H} \mathbb{E}_{\substack{s\sim\rho_{\pi_{k,h}} \\ a\sim\pi_{k-1,h}}} \left[ -\rho(\theta_h) A_h^{\pi_{k-1}}(s,a) \right] + \sum_{i}^{m} \lambda_{t,i} w_{t,i}(\boldsymbol{\pi}_k) + \frac{\beta}{2}\sum_{i}^{m} w_{t,i}^2(\boldsymbol{\pi}_k) \\
&= \sum_{h=t}^{H} \mathbb{E}_{\substack{s\sim\rho_{\pi_{k,h}} \\ a\sim\pi_{k-1,h}}} \left[ -\rho(\theta_h) A_h^{\pi_{k-1}}(s,a) \right] + \sum_{i=1}^{m} \lambda_{t,i}\left(\max\left\{0, \frac{\lambda_{t,i}}{\beta} + \Psi_{C_i,t}(\boldsymbol{\pi}_{k-1}, \boldsymbol{\pi}_k)\right\} - \frac{\lambda_{t,i}}{\beta}\right) \\
&\quad + \frac{\beta}{2}\sum_{i=1}^{m}\left(\max\left\{0, \frac{\lambda_{t,i}}{\beta} + \Psi_{C_i,t}(\boldsymbol{\pi}_{k-1}, \boldsymbol{\pi}_k)\right\} - \frac{\lambda_{t,i}}{\beta}\right)^2 \\
&= \sum_{h=t}^{H} \mathbb{E}_{\substack{s\sim\rho_{\pi_{k,h}} \\ a\sim\pi_{k-1,h}}} \left[ -\rho(\theta_h) A_h^{\pi_{k-1}}(s,a) \right] + \frac{\beta}{2}\sum_{i=1}^{m}\left(\max\left\{0, \frac{\lambda_{t,i}}{\beta} + \Psi_{C_i,t}(\boldsymbol{\pi}_{k-1}, \boldsymbol{\pi}_k)\right\}^2 - \frac{\lambda_{t,i}^2}{\beta^2}\right)
\end{aligned}$$

Hence, we finally get

$$\begin{aligned}
(\pi_{k,t}^{\star}, \boldsymbol{\lambda}_t^{\star}) &= \max_{\boldsymbol{\lambda}\geqslant 0} \min_{\pi_{k,t}} \mathcal{L}_t(\boldsymbol{\pi}_k, \boldsymbol{\lambda}, \beta) \\
&= \max_{\boldsymbol{\lambda}\geqslant 0} \min_{\pi_{k,t}} \sum_{h=t}^{H} \mathbb{E}_{\substack{s\sim\rho_{\pi_{k,h}} \\ a\sim\pi_{k-1,h}}} \left[ -\rho(\theta_h) A_h^{\pi_{k-1}}(s,a) \right] + \frac{\beta}{2}\sum_{i=1}^{m}\left(\max\left\{0, \Psi_{C_i,t}(\boldsymbol{\pi}_{k-1}, \boldsymbol{\pi}_k) + \frac{\lambda_{t,i}}{\beta}\right\}^2 - \frac{\lambda_{t,i}^2}{\beta^2}\right)
\end{aligned}$$

$\square$

**Lemma A.3.** *Consider two problems, Problem (P) and Problem (Q) below. For sufficiently large $\lambda_i > \bar{\lambda} \ \forall \ i$ and $\beta > \bar{\beta}$ for some finite $\bar{\lambda}$ and finite $\bar{\beta}$, the optimal solution set of Problem (Q) (equivalent version of Problem (6)) is identical to the optimal solution set of Problem (P).*

*Problem (P) :*

$$\mathcal{L}_t^P(\boldsymbol{\pi}_k, \boldsymbol{\lambda}, \boldsymbol{x}, \beta) := \sum_{h=t}^{H} \mathbb{E}_{\substack{s\sim\rho_{\pi_{k,h}} \\ a\sim\pi_{k-1,h}}} \left[ -\rho(\theta_h) A_h^{\pi_{k-1}}(s,a) \right] + \sum_{i}^{m} \lambda_{t,i} w_{t,i}(\boldsymbol{\pi}_k) + \frac{\beta}{2}\sum_{i}^{m} w_{t,i}^2(\boldsymbol{\pi}_k)$$

$$\text{Then,} \qquad (\pi_{k,t}^{\star}, \boldsymbol{\lambda}_t^{\star}, \boldsymbol{x}_t^{\star}) = \max_{\boldsymbol{\lambda}\geqslant 0} \min_{\pi_{k,t},\boldsymbol{x}} \mathcal{L}_t^P(\boldsymbol{\pi}_k, \boldsymbol{\lambda}, \boldsymbol{x}, \beta) \tag{P}$$

*Problem (Q) :*

$$\mathcal{L}_t^Q(\boldsymbol{\pi}_k, \boldsymbol{\lambda}, \boldsymbol{x}, \beta) := \sum_{h=t}^{H} \mathop{\mathbb{E}}_{\substack{s \sim \rho_{\pi_{k,h}} \\ a \sim \pi_{k-1,h}}} \left[ -\rho(\theta_h) A_h^{\boldsymbol{\pi}_{k-1}}(s,a) \right] + \sum_i^m \lambda_{t,i} \Psi_{C_i,t}^+(\boldsymbol{\pi}_{k-1}, \boldsymbol{\pi}_k, \theta)$$

$$+ \frac{\beta}{2} \sum_i^m \Psi_{C_i,t}^+(\boldsymbol{\pi}_{k-1}, \boldsymbol{\pi}_k, \theta)^2$$

*Then,* $\qquad (\boldsymbol{\pi}_{k,t}^\star, \boldsymbol{\lambda}_t^\star, \boldsymbol{x}_t^\star) = \max_{\boldsymbol{\lambda} \geqslant 0} \min_{\boldsymbol{\pi}_{k,t}, \boldsymbol{x}} \mathcal{L}_t^Q(\boldsymbol{\pi}_k, \boldsymbol{\lambda}, \boldsymbol{x}, \beta)$ $\qquad\qquad$ (Q)

*, where* $x^+ := \max(0, x)$, *and*

$$\Psi_{C_i,t}(\boldsymbol{\pi}_{k-1}, \boldsymbol{\pi}_k, \theta) := \sum_{h=t}^{H} \mathop{\mathbb{E}}_{\substack{s \sim \rho_{\pi_{k,h}} \\ a \sim \pi_{k-1,h}}} \left[ \rho(\theta_h) A_{C_i,h}^{\boldsymbol{\pi}_{k-1}}(s,a) \right] + (J_{C_i}(\boldsymbol{\pi}_{k-1}) - d_i).$$

*Proof.* This proof uses some ideas given in [41] for Part 1 below.

**Part 1** - Solution of Problem (P) is solution of Problem (Q).

Suppose $\bar{\theta}_t$ is the optimum of the constrained Problem (P) augmented with the quadratic penalty. Let $\bar{\lambda}_t$ be the corresponding Lagrange multiplier vector for its dual problem, and $\bar{\beta}$ be the additive quadratic penalty coefficient. Then for $\lambda_{t,i} \geqslant ||\bar{\lambda}||_\infty \ \forall i$ and $\beta \geqslant ||\bar{\beta}||_\infty$, $\bar{\theta}$ is also a minimizer of its ReLU-penalized optimization Problem (Q) as below. Let $\Omega(\theta_t) := \sum_{h=t}^{H} \mathop{\mathbb{E}}_{\substack{s \sim \rho_{\pi_{k,h}} \\ a \sim \pi_{k-1,h}}} \left[ -\rho(\theta_h) A_h^{\boldsymbol{\pi}_{k-1}}(s,a) \right]$. Then it follows that:

$$\Omega(\theta_t) + \sum_i^m \lambda_{t,i} \Psi_{C_i,t}^+(\boldsymbol{\pi}_{k-1}, \boldsymbol{\pi}_k, \theta) + \frac{\beta}{2} \sum_i^m \Psi_{C_i,t}^+(\boldsymbol{\pi}_{k-1}, \boldsymbol{\pi}_k, \theta)^2 \geqslant \Omega(\theta_t) + \sum_i^m \bar{\lambda}_i \Psi_{C_i,t}^+(\boldsymbol{\pi}_{k-1}, \boldsymbol{\pi}_k, \theta) + \frac{\bar{\beta}}{2} \sum_i^m \Psi_{C_i,t}^+(\boldsymbol{\pi}_{k-1}, \boldsymbol{\pi}_k, \theta)^2$$

$$\geqslant \Omega(\theta_t) + \sum_i^m \bar{\lambda}_i \Psi_{C_i,t}(\boldsymbol{\pi}_{k-1}, \boldsymbol{\pi}_k, \theta) + \frac{\bar{\beta}}{2} \sum_i^m \Psi_{C_i,t}(\boldsymbol{\pi}_{k-1}, \boldsymbol{\pi}_k, \theta)^2$$

By assumption, $\bar{\theta}_t$ is a Karush-Kuhn-Tucker point in the constrained Problem (P), at which KKT conditions are satisfied with the Lagrange multiplier vector $\bar{\lambda}$ and $\bar{\beta}$. We then have:

$$\Omega(\theta_t) + \sum_i^m \bar{\lambda}_i \Psi_{C_i,t}(\boldsymbol{\pi}_{k-1}, \boldsymbol{\pi}_k, \theta) + \frac{\bar{\beta}}{2} \sum_i^m \Psi_{C_i,t}(\boldsymbol{\pi}_{k-1}, \boldsymbol{\pi}_k, \theta)^2 \geqslant \Omega(\bar{\theta}_t) + \sum_i^m \bar{\lambda}_i \Psi_{C_i,t}(\boldsymbol{\pi}_{k-1}, \boldsymbol{\pi}_k, \bar{\theta}) + \frac{\bar{\beta}}{2} \sum_i^m \Psi_{C_i,t}(\boldsymbol{\pi}_{k-1}, \pi, \bar{\theta})^2$$

$$= \Omega(\bar{\theta}_t) + \sum_i^m \bar{\lambda}_i \Psi_{C_i,t}^+(\boldsymbol{\pi}_{k-1}, \boldsymbol{\pi}_k, \bar{\theta}) + \frac{\bar{\beta}}{2} \sum_i^m \Psi_{C_i,t}^+(\boldsymbol{\pi}_{k-1}, \boldsymbol{\pi}_k, \bar{\theta})^2$$

$$= \Omega(\bar{\theta}_t) + \sum_i^m \lambda_{t,i} \Psi_{C_i,t}^+(\boldsymbol{\pi}_{k-1}, \boldsymbol{\pi}_k, \bar{\theta}) + \frac{\beta}{2} \sum_i^m \Psi_{C_i,t}^+(\boldsymbol{\pi}_{k-1}, \boldsymbol{\pi}_k, \bar{\theta})^2$$

, where the first line holds because $\bar{\theta}_t$ minimizes the Lagrange function, and the second line is derived from the complementary slackness. Thus, we conclude that for the objective function of Problem (Q), call it $\mathcal{L}^Q(\theta_t)$, we have $\mathcal{L}^Q(\theta_t) \geqslant \mathcal{L}^Q(\bar{\theta}_t)$ for all $\theta_t \in \Theta$, which means $\bar{\theta}_t$ is a minimizer of the quadratic damped optimization Problem (Q).

**Part 2** - Solution of Problem (Q) is solution of Problem (P).

Let $\widetilde{\theta}_t$ be an optimal point of the quadratic damped Problem (Q), with $\bar{\theta}_t$ and $\bar{\lambda}$ being the same as defined above. Then, if $\widetilde{\theta}_t$ is in the set of feasible solutions $S_{\text{feasible}} = \{\theta \mid \Psi_{C_i,t}(\boldsymbol{\pi}_{k-1}, \boldsymbol{\pi}_k, \theta) \leqslant 0 \ \ \forall i\}$, we have:

$$\Omega(\widetilde{\theta}_t) = \Omega(\widetilde{\theta}_t) + \sum_i^m \lambda_{t,i} \Psi_{C_i,t}^+(\boldsymbol{\pi}_{k-1}, \boldsymbol{\pi}_k, \widetilde{\theta}) + \frac{\beta}{2} \sum_i^m \Psi_{C_i,t}^+(\boldsymbol{\pi}_{k-1}, \boldsymbol{\pi}_k, \widetilde{\theta})$$

$$\leqslant \Omega(\theta_t) + \sum_i^m \lambda_{t,i} \Psi_{C_i,t}^+(\boldsymbol{\pi}_{k-1}, \boldsymbol{\pi}_k, \theta) + \frac{\beta}{2} \sum_i^m \Psi_{C_i,t}^+(\boldsymbol{\pi}_{k-1}, \boldsymbol{\pi}_k, \theta)$$

$$= \Omega(\theta_t)$$

The inequality above indicates $\widetilde{\theta}_t$ is also optimal in the constrained Problem (P). Now, if $\bar{\theta}$ is not feasible, we have:

$$\Omega(\bar{\theta}_t) + \sum_i^m \lambda_{t,i}\Psi_{C_i,t}^+(\boldsymbol{\pi}_{k-1},\boldsymbol{\pi}_k,\bar{\theta}) + \frac{\beta}{2}\sum_i^m \Psi_{C_i,t}^+(\boldsymbol{\pi}_{k-1},\boldsymbol{\pi}_k,\bar{\theta})^2 = \Omega(\bar{\theta}_t) + \sum_i^m \bar{\lambda}_i\Psi_{C_i,t}^+(\boldsymbol{\pi}_{k-1},\boldsymbol{\pi}_k,\bar{\theta}) + \frac{\bar{\beta}}{2}\sum_i^m \Psi_{C_i,t}^+(\boldsymbol{\pi}_{k-1},\boldsymbol{\pi}_k,\bar{\theta})^2$$

$$= \Omega(\bar{\theta}_t) + \sum_i^m \bar{\lambda}_i\Psi_{C_i,t}(\boldsymbol{\pi}_{k-1},\boldsymbol{\pi}_k,\bar{\theta}) + \frac{\bar{\beta}}{2}\sum_i^m \Psi_{C_i,t}(\boldsymbol{\pi}_{k-1},\boldsymbol{\pi}_k,\bar{\theta})^2$$

$$\leqslant \Omega(\widetilde{\theta}_t) + \sum_i^m \bar{\lambda}_i\Psi_{C_i,t}(\boldsymbol{\pi}_{k-1},\boldsymbol{\pi}_k,\widetilde{\theta}) + \frac{\bar{\beta}}{2}\sum_i^m \Psi_{C_i,t}(\boldsymbol{\pi}_{k-1},\boldsymbol{\pi}_k,\widetilde{\theta})^2$$

$$\leqslant \Omega(\widetilde{\theta}_t) + \sum_i^m \bar{\lambda}_i\Psi_{C_i,t}^+(\boldsymbol{\pi}_{k-1},\boldsymbol{\pi}_k,\widetilde{\theta}) + \frac{\bar{\beta}}{2}\sum_i^m \Psi_{C_i,t}^+(\boldsymbol{\pi}_{k-1},\boldsymbol{\pi}_k,\widetilde{\theta})^2$$

$$\leqslant \Omega(\widetilde{\theta}_t) + \sum_i^m \lambda_{t,i}\Psi_{C_i,t}^+(\boldsymbol{\pi}_{k-1},\boldsymbol{\pi}_k,\widetilde{\theta}) + \frac{\beta}{2}\sum_i^m \Psi_{C_i,t}^+(\boldsymbol{\pi}_{k-1},\boldsymbol{\pi}_k,\widetilde{\theta})^2$$

, which is a contradiction to the assumption that $\widetilde{\theta}_t$ is a minimizer of the penalized optimization Problem (Q). Thus, $\widetilde{\theta}_t$ can only be the feasible optimal solution for Problem (P).

$\square$

**Lemma A.4.** *Consider two problems, Problem* (P') *and Problem* (R). *For sufficiently large* $\beta > \bar{\beta}$ *for some finite* $\bar{\beta}$, *the feasible optimal solution set of Problem* (R) *(equivalent version of Problem* (3)*) is identical to the solution set of Problem* (P')*.*

*Problem* (P') *:*

$$\mathcal{L}_t^{P'}(\boldsymbol{\pi}_k,\boldsymbol{\lambda},\boldsymbol{x},\beta) := \sum_{h=t}^{H} \mathop{\mathbb{E}}_{\substack{s\sim\rho_{\pi_{k,h}} \\ a\sim\pi_{k-1,h}}} \left[ -\rho(\theta_h)A_h^{\boldsymbol{\pi}_{k-1}}(s,a) \right] + \sum_i^m \lambda_{t,i}w_{t,i}(\boldsymbol{\pi}_k) + \frac{\beta}{2}\sum_i^m w_{t,i}^2(\boldsymbol{\pi}_k)$$

$$\text{Then,} \qquad (\pi_{k,t}^\star,\boldsymbol{\lambda}_t^\star,\boldsymbol{x}_t^\star) = \max_{\boldsymbol{\lambda}\geqslant 0}\min_{\pi_{k,t},\boldsymbol{x}} \mathcal{L}_t^{P'}(\boldsymbol{\pi}_k,\boldsymbol{\lambda},\boldsymbol{x},\beta) \qquad \text{(P')}$$

*Problem* (R) *:*

$$\mathcal{L}_t^R(\boldsymbol{\pi}_k,\boldsymbol{\lambda},\boldsymbol{x}) := \sum_{h=t}^{H} \mathop{\mathbb{E}}_{\substack{s\sim\rho_{\pi_{k,h}} \\ a\sim\pi_{k-1,h}}} \left[ -\rho(\theta_h)A_h^{\boldsymbol{\pi}_{k-1}}(s,a) \right] + \sum_i^m \lambda_{t,i}w_{t,i}(\boldsymbol{\pi}_k)$$

$$\text{Then,} \qquad (\pi_{k,t}^\star,\boldsymbol{\lambda}_t^\star,\boldsymbol{x}_t^\star) = \max_{\boldsymbol{\lambda}\geqslant 0}\min_{\pi_{k,t},\boldsymbol{x}} \mathcal{L}_t^R(\boldsymbol{\pi}_k,\boldsymbol{\lambda},\boldsymbol{x}) \qquad \text{(R)}$$

*Proof.* Recall that we are using parameterized policies, hence we overload notation as $\theta \equiv \pi$ frequently. For brevity, denote $\Omega_t(\pi) := \sum_{h=t}^{H} \mathop{\mathbb{E}}_{\substack{s\sim\rho_{\pi_{k,h}} \\ a\sim\pi_{k-1,h}}} \left[ \rho(\theta_h)A_h^{\boldsymbol{\pi}_{k-1}}(s,a) \right]$. We will also go back and forth between the equivalent problems of Problem (P') and Problem (6) of the main paper in Section 3.

**Part 1.** Solution of Problem (R) is solution of Problem (P').

Suppose that $\pi^\star$ is the optimal feasible policy for the primal Problem (R), which is a Lagrangian version of Problem (3). Consider the corresponding Langrangian dual parameter $\boldsymbol{\lambda}^\star$ of $\pi^\star$, which satisfies the KKT conditon,

$$\nabla_\pi \mathcal{L}_t^R\left(\pi_{k,t}^\star,\boldsymbol{\lambda}^\star,\boldsymbol{x}^\star\right) = -\nabla_\pi\Omega_t\left(\pi_{k,t}^\star\right) + \sum_{i=1}^m \lambda_{t,i}^*\nabla_\pi w_{t,i}\left(\boldsymbol{\pi}_k^\star\right) = 0$$

and the second-order sufficient condition that for all non-zero vectors $\boldsymbol{u}$ that satisfy $\boldsymbol{u}^T\nabla_\pi w_{t,i}\left(\boldsymbol{\pi}_k^\star\right) = 0$, we have

$$\boldsymbol{u}^T\nabla_\pi^2\mathcal{L}_t^R\left(\pi_{k,t}^\star,\boldsymbol{\lambda}^\star,\boldsymbol{x}^\star\right)\boldsymbol{u} > 0 \qquad \text{(A)}$$

Compare Equation (R) and Equation (P'), we have,

$$\nabla_\pi \mathcal{L}_t^{P'}\left(\pi_{k,t}^\star, \boldsymbol{\lambda}^\star, \boldsymbol{x}^\star, \beta\right) = -\nabla_\pi \Omega_t\left(\pi_{k,t}^\star\right) + \sum_{i=1}^m \lambda_{t,i}^\star \nabla_\pi w_{t,i}\left(\boldsymbol{\pi}_k^\star\right) + \beta \sum_{i=1}^m w_{t,i}\left(\boldsymbol{\pi}_k^\star\right) \nabla_\pi w_{t,i}\left(\boldsymbol{\pi}_k^\star\right)$$

$$= \nabla_\pi \mathcal{L}_t^R\left(\pi_{k,t}^\star, \boldsymbol{\lambda}^\star, \boldsymbol{x}^\star\right) + \beta \sum_{i=1}^m w_{t,i}\left(\boldsymbol{\pi}_k^\star\right) \nabla_\pi w_{t,i}\left(\boldsymbol{\pi}_k^\star\right)$$

$$= 0$$

, where we use $w_{t,i}(\boldsymbol{\pi}_k^\star) := \Psi_{C_i,t}(\boldsymbol{\pi}_{k-1}, \boldsymbol{\pi}^\star) + x_{t,i}^\star = 0$ with the feasible policy $\boldsymbol{\pi}^\star$. Moreover,

$$\nabla_\pi^2 \mathcal{L}_t^{P'}\left(\pi_{k,t}^\star, \boldsymbol{\lambda}^\star, \boldsymbol{x}^\star, \beta\right) = -\nabla_\pi^2 \Omega_t\left(\pi_{k,t}^\star\right)$$

$$+ \sum_{i=1}^m \lambda_{t,i}^\star \nabla_\pi^2 w_{t,i}\left(\boldsymbol{\pi}_k^\star\right) + \beta \nabla_\pi \boldsymbol{w}_t\left(\boldsymbol{\pi}_k^\star\right) \nabla_\pi \boldsymbol{w}_t\left(\boldsymbol{\pi}_k^\star\right)^T$$

$$= \nabla_\pi^2 \mathcal{L}_t^R\left(\pi_{k,t}^\star, \boldsymbol{\lambda}^\star, \boldsymbol{x}^\star\right) + \beta \nabla_\pi \boldsymbol{w}_t\left(\boldsymbol{\pi}_k^\star\right) \nabla_\pi \boldsymbol{w}_t\left(\boldsymbol{\pi}_k^\star\right)^T.$$

To prove that $(\pi_{k,t}^\star, \boldsymbol{\lambda}^\star)$ is a strict minimum solution to $\mathcal{L}_t^{P'}(\boldsymbol{\pi}_k, \boldsymbol{\lambda}, \boldsymbol{x}, \beta)$, we only need to prove the following is true for sufficiently large $\beta$,

$$\nabla_\pi^2 \mathcal{L}_t^{P'}\left(\boldsymbol{\pi}_k^\star, \boldsymbol{\lambda}^\star, \boldsymbol{x}^\star, \beta\right) > 0.$$

If the above is not true, then for any large $\beta$, there exists $\boldsymbol{u}_t$ such that $\|\boldsymbol{u}_t\| = 1$ and satisfies

$$\boldsymbol{u}_t^T \nabla_\pi^2 \mathcal{L}_t^{P'}\left(\boldsymbol{\pi}_k^\star, \boldsymbol{\lambda}^\star, \boldsymbol{x}^\star, \beta\right) \boldsymbol{u}_t = \boldsymbol{u}_t^T \nabla_\pi^2 \mathcal{L}_t^R\left(\boldsymbol{\pi}_k^\star, \boldsymbol{\lambda}^\star, \boldsymbol{x}^\star\right) \boldsymbol{u}_t + \beta \left\| \nabla_\pi \boldsymbol{w}_t\left(\boldsymbol{\pi}_k^\star\right)^T \boldsymbol{u}_t \right\|^2 \leqslant 0$$

$$\Rightarrow \left\| \nabla_\pi \boldsymbol{w}_t\left(\boldsymbol{\pi}_k^\star\right)^T \boldsymbol{u}_t \right\|^2 \leqslant -\frac{1}{\beta} \boldsymbol{u}_t^T \nabla_\pi^2 \mathcal{L}_t^R\left(\boldsymbol{\pi}_k^\star, \boldsymbol{\lambda}^\star, \boldsymbol{x}^\star\right) \boldsymbol{u}_t \to 0, \text{ as } \beta \to \infty.$$

Therefore, $\{\boldsymbol{u}_h\}$ is a bounded sequence and there must be a limit point, denoted by $\mathring{\boldsymbol{u}}$. Then

$$\nabla_\pi \boldsymbol{w}_t\left(\boldsymbol{\pi}_k^\star\right)^T \mathring{\boldsymbol{u}} = 0$$
$$\mathring{\boldsymbol{u}}^T \nabla_\pi^2 \mathcal{L}_t^R\left(\boldsymbol{\pi}_k^\star, \boldsymbol{\lambda}^\star, \boldsymbol{x}^\star\right) \mathring{\boldsymbol{u}} \leqslant 0.$$

The above contradicts Equation (A), so the conclusion. Hence, $\pi_{k,t}^\star$ is also the optimal feasible policy for the primal-dual Problem (P').

**Part 2.** Solution of Problem (P') is solution of Problem (R).

This part is straightforward since it is a standard result. Please see Chapter 2 and Chapter 9 of [9], and Chapter 2 and Chapter 4 of [8] for the proof. For completeness, we provide the result below.

Suppose $\pi_{k,t}^\star$ in the feasible optimal solution set of the primal-dual Problem (P'). Let $\boldsymbol{\lambda}^\star$ be the corresponding dual parameter of $\pi_{k,t}^\star$. Consider Problem (6), which is an equivalent version of Problem (P'). For any feasible $\boldsymbol{\pi}_k$, we have

$$\mathcal{L}_t\left(\boldsymbol{\pi}_k^\star, \boldsymbol{\lambda}^\star, \beta\right) \leqslant \mathcal{L}_t\left(\boldsymbol{\pi}_k, \boldsymbol{\lambda}^\star, \beta\right).$$

Now we have two cases:

Case 1. When $\frac{\lambda_{t,i}^\star}{\beta} + \Psi_{C_i,t}(\boldsymbol{\pi}_{k-1}, \boldsymbol{\pi}_k^\star) > 0$, we have

$$\mathcal{L}_t\left(\boldsymbol{\pi}_k, \boldsymbol{\lambda}^\star, \beta\right) = -\Omega_t(\pi_{k,t}) + \sum_{i=1}^m \lambda_{t,i}^\star \Psi_{C_i,t}(\boldsymbol{\pi}_{k-1}, \boldsymbol{\pi}_k) + \frac{\beta}{2} \sum_{i=1}^m \Psi_{C_i,t}^2(\boldsymbol{\pi}_{k-1}, \boldsymbol{\pi}_k)$$

$$= -\Omega_t(\pi_{k,t}) + \sum_{i=1}^m \beta \Psi_{C_i,t}(\boldsymbol{\pi}_{k-1}, \boldsymbol{\pi}_k)\left(\frac{\lambda_{t,i}^\star}{\beta} + \Psi_{C_i,t}(\boldsymbol{\pi}_{k-1}, \boldsymbol{\pi}_k)\right) - \frac{\beta}{2} \sum_{i=1}^m \Psi_{C_i,t}^2(\boldsymbol{\pi}_{k-1}, \boldsymbol{\pi}_k)$$

$$\leqslant -\Omega_t(\pi_{k,t})$$

where the last step uses $\beta > 0$, $\Psi_{C_i,t}(\boldsymbol{\pi}_{k-1}, \boldsymbol{\pi}_k) < 0$, and $\frac{\lambda_{t,i}^\star}{\beta} + \Psi_{C_i,t}(\boldsymbol{\pi}_{k-1}, \boldsymbol{\pi}_k) > 0$.

Case 2. When $\frac{\lambda_{t,i}^\star}{\beta} + \Psi_{C_i,t}(\boldsymbol{\pi}_{k-1}, \boldsymbol{\pi}_k^\star) \leqslant 0$, we have

$$\mathcal{L}_t\left(\boldsymbol{\pi}_k, \boldsymbol{\lambda}^\star, \beta\right) = -\Omega_t(\pi_{k,t}) - \frac{1}{2\beta} \sum_{i=1}^m \lambda_{t,i}^{\star 2} \leqslant -\Omega_t(\pi_{k,t}).$$

Now, combining both cases above, we have $\mathcal{L}_t\left(\boldsymbol{\pi}_k, \boldsymbol{\lambda}^\star, \beta\right) \leqslant -\Omega_t(\pi_{k,t})$. On the other hand, $\mathcal{L}_t\left(\boldsymbol{\pi}_k^\star, \boldsymbol{\lambda}^\star, \beta\right) = \mathcal{L}_t(\boldsymbol{\pi}_k^\star, \boldsymbol{\lambda}^\star, \boldsymbol{x}^\star, \beta) = -\Omega_t(\pi_{k,t}^\star)$. Thus, the combining all of the above we get

$$-\Omega_t(\pi_{k,t}^\star) = \mathcal{L}_t\left(\boldsymbol{\pi}_k^\star, \boldsymbol{\lambda}^\star, \beta\right) \leqslant \mathcal{L}_t\left(\boldsymbol{\pi}_k, \boldsymbol{\lambda}^\star, \beta\right) \leqslant -\Omega_t(\pi_{k,t}).$$

$\square$

**Theorem 3.3.** *Let $\pi^{(3)\star}$ be a solution to Problem* (3), *and let $\left(\pi^{(6)\star}, \boldsymbol{\lambda}^{(6)\star}\right)$ be a solution to Problem* (6). *Then, for sufficiently large $\beta > \bar{\beta}$ and $\lambda_{t,i} > \bar{\lambda}$ $\forall i$, $\pi^{(3)\star}$ is a solution to Problem* (6), *and $\pi^{(6)\star}$ is a solution to Problem* (3).

*Proof.* We prove this result as a two-step process.

First, we show that the solution sets of the below problems are identical. See Lemma A.3 for the proof.

Problem (P).

$$\mathcal{L}_t^P(\boldsymbol{\pi}_k, \boldsymbol{\lambda}, \boldsymbol{x}, \beta) := \sum_{h=t}^H \mathop{\mathbb{E}}_{\substack{s \sim \rho_{\pi_{k,h}} \\ a \sim \pi_{k-1,h}}} \left[ -\rho(\theta_h) A_h^{\boldsymbol{\pi}_{k-1}}(s,a)\right] + \sum_i^m \lambda_{t,i} w_{t,i}(\boldsymbol{\pi}_k) + \frac{\beta}{2}\sum_i^m w_{t,i}^2(\boldsymbol{\pi}_k)$$

$$\text{Then,} \qquad (\pi_{k,t}^\star, \boldsymbol{\lambda}_t^\star, \boldsymbol{x}_t^\star) = \max_{\boldsymbol{\lambda} \geqslant 0} \min_{\pi_{k,t}, \boldsymbol{x}} \mathcal{L}_t^P(\boldsymbol{\pi}_k, \boldsymbol{\lambda}, \boldsymbol{x}, \beta) \tag{P}$$

Problem (Q).

$$\mathcal{L}_t^Q(\boldsymbol{\pi}_k, \boldsymbol{\lambda}, \boldsymbol{x}, \beta) := \sum_{h=t}^H \mathop{\mathbb{E}}_{\substack{s \sim \rho_{\pi_{k,h}} \\ a \sim \pi_{k-1,h}}} \left[ -\rho(\theta_h) A_h^{\boldsymbol{\pi}_{k-1}}(s,a)\right] + \sum_i^m \lambda_{t,i} \Psi_{C_i,t}^+(\boldsymbol{\pi}_{k-1}, \boldsymbol{\pi}_k, \theta)$$

$$+ \frac{\beta}{2}\sum_i^m \Psi_{C_i,t}^+(\boldsymbol{\pi}_{k-1}, \boldsymbol{\pi}_k, \theta)^2$$

$$\text{Then,} \qquad (\pi_{k,t}^\star, \boldsymbol{\lambda}_t^\star, \boldsymbol{x}_t^\star) = \max_{\boldsymbol{\lambda} \geqslant 0} \min_{\pi_{k,t}, \boldsymbol{x}} \mathcal{L}_t^Q(\boldsymbol{\pi}_k, \boldsymbol{\lambda}, \boldsymbol{x}, \beta) \tag{Q}$$

Second, we show that the solution sets of the below problems are identical. See Lemma A.4 for the proof.

Problem (P') :

$$\mathcal{L}_t^{P'}(\boldsymbol{\pi}_k, \boldsymbol{\lambda}, \boldsymbol{x}, \beta) := \sum_{h=t}^H \mathop{\mathbb{E}}_{\substack{s \sim \rho_{\pi_{k,h}} \\ a \sim \pi_{k-1,h}}} \left[ -\rho(\theta_h) A_h^{\boldsymbol{\pi}_{k-1}}(s,a)\right] + \sum_i^m \lambda_{t,i} w_{t,i}(\boldsymbol{\pi}_k) + \frac{\beta}{2}\sum_i^m w_{t,i}^2(\boldsymbol{\pi}_k)$$

$$\text{Then,} \qquad (\pi_{k,t}^\star, \boldsymbol{\lambda}_t^\star, \boldsymbol{x}_t^\star) = \max_{\boldsymbol{\lambda} \geqslant 0} \min_{\pi_{k,t}, \boldsymbol{x}} \mathcal{L}_t^{P'}(\boldsymbol{\pi}_k, \boldsymbol{\lambda}, \boldsymbol{x}, \beta) \tag{P'}$$

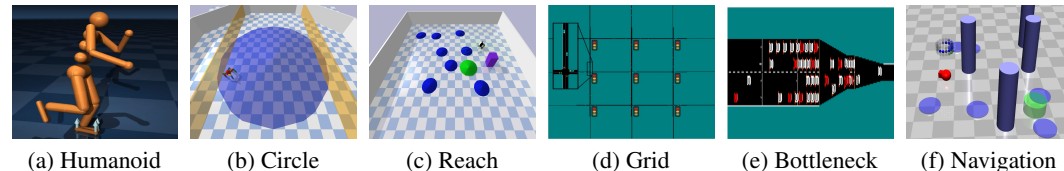

| (a) Humanoid | (b) Circle | (c) Reach | (d) Grid | (e) Bottleneck | (f) Navigation |

Figure 5: The Humanoid, Circle, Reach, Grid, Bottleneck, and Navigation tasks. (a) **Humanoid**: The agent is to run as fast as possible on a flat surface, while not exceeding a specified speed limit i.e. the cost constraint. (b) **Circle**: The agent is rewarded for moving in a specified circle but is penalized if the diameter of the circle is larger than some value [3]. (c) **Reach**: The agent is rewarded for reaching a goal while avoiding obstacles (cost constraints) that are placed to hinder the agent [30]. (d) **Grid**: The agent controls traffic lights in a 3x3 road network and is rewarded for high traffic throughput but is constrained to let lights be red for at most 5 consecutive seconds [37]. (e) **Bottleneck**: The agent controls vehicles (red) in a merging traffic situation and is rewarded for maximizing the number of vehicles that pass through but is constrained to ensure that white vehicles (not controlled by agent) have "low" speed for no more than 10 seconds [37]. (f) **Navigation**: The agent is rewarded for reaching the target area (green) but is constrained to avoid hazards (light purple) and impassible pillars (dark purple). The cost for hazards and pillars is different [30].

Problem (R) :

$$\mathcal{L}_t^R(\boldsymbol{\pi}_k, \boldsymbol{\lambda}, \boldsymbol{x}) := \sum_{\substack{h=t}}^{H} \mathbb{E}_{\substack{s \sim \rho_{\pi_{k,h}} \\ a \sim \pi_{k-1,h}}} \big[ -\rho(\theta_h) A_h^{\boldsymbol{\pi}_{k-1}}(s,a) \big] + \sum_i^m \lambda_{t,i} w_{t,i}(\boldsymbol{\pi}_k)$$

$$\text{Then,} \qquad (\pi_{k,t}^\star, \boldsymbol{\lambda}_t^\star, \boldsymbol{x}_t^\star) = \max_{\boldsymbol{\lambda} \geqslant 0} \min_{\pi_{k,t}, \boldsymbol{x}} \mathcal{L}_t^R(\boldsymbol{\pi}_k, \boldsymbol{\lambda}, \boldsymbol{x}) \tag{R}$$

Now, it follows from equivalency that the optimal solution of Problem (Q) and Problem (R), and hence Problem (6) and Problem (3), is the same.

$\square$

## A.2 Experiments Revisited

Below we detail the experimental attributes that we used in benchmarking. See Figure 5 for the environment details. All our experiments are run in the `omnisafe` module [22].

### A.2.1 Environment Details

Comprehensively, our experiments consist of eight tasks ranging from more superficial (Run and Circle tasks) to relatively more stochastic and sophisticated (Bottleneck and Grid tasks), each training different robots. They come from three well-known safe RL benchmark environments, Safe MuJoCo, Bullet-Safety-Gym, and Safety-Gym. For agents maneuvering on a two-dimensional plane, the cost is calculated as $C(s,a) = \sqrt{v_x^2 + v_y^2}$. For agents moving along a straight line, the cost is calculated as $C(s,a) = |v_x|$, where $v_x$ and $v_y$ are the velocities of the agent in the x and y directions.

**Circle** This environment is inspired by [3]. Reward is maximized by moving along a circle of radius $d$:

$$R = \frac{v^{\mathrm{T}}[-y, x]}{1 + \left|\sqrt{x^2 + y^2} - d\right|},$$

but the safety region $x_{\text{lim}}$ is smaller than the radius $d : C = \mathbf{1}[x > x_{\text{lim}}]$.

**Navigation** This environment is inspired by [30]. Reward is maximized by getting close to the destination $R = \text{Dist}(target, s_{t-1}) - \text{Dist}(target, s_t)$, but it yields a cost of +1 when the agent hits the hazard or the pillar. The two different types of cost functions are returned separately and have different thresholds. In out setting, $d_1 = 25$ for the hazard constraint and $d_2 = 20$ for the pillar constraint.

Since the main goal in MuJoCo is to train the robot to locomote on the plane, we call it the "Run" task in our article. Our chosen two robots are the relatively complex types in MuJoCo: Ant and Humanoid. **OpenAI Gym** is open source at https://github.com/openai/gym, and has a documentation

| Hyperparameter | APPO | PDO | FOCOPS | CPPO-PID | IPO | P3O | CPO | TRPO-L | PCPO |
|---|---|---|---|---|---|---|---|---|---|
| Actor Net layers | (32, 32) | (32, 32) | (32, 32) | (32, 32) | (32, 32) | (32, 32) | (32, 32) | (32, 32) | (32, 32) |
| Critic Net layers | (32, 32) | (32, 32) | (32, 32) | (32, 32) | (32, 32) | (32, 32) | (32, 32) | (32, 32) | (32, 32) |
| Activation | tanh | tanh | tanh | tanh | tanh | tanh | tanh | tanh | tanh |
| Initial log std | 0.5 | 0.5 | 0.5 | 0.5 | 0.5 | 0.5 | 0.5 | 0.5 | 0.5 |
| Discount $\gamma$ | 0.99 | 0.95 | 0.995 | 0.995 | 0.99 | 0.99 | 0.99 | 0.99 | 0.99 |
| Policy lr | $3e-4$ | $3e-4$ | $3e-4$ | $3e-4$ | $3e-4$ | $3e-4$ | $3e-4$ | $3e-4$ | $3e-4$ |
| Critic Net lr | $1e-3$ | $1e-3$ | $1e-3$ | $1e-3$ | $1e-3$ | $1e-3$ | $1e-3$ | $1e-3$ | $1e-3$ |
| No. of episodes | 500 | 500 | 500 | 500 | 500 | 500 | 500 | 500 | 500 |
| Steps per epochs | 300 | 300 | 300 | 300 | 300 | 300 | 300 | 300 | 300 |
| Target KL | 0.01 | 0.01 | 0.01 | 0.01 | 0.01 | 0.01 | 0.01 | 0.01 | 0.01 |
| KL early stop | True | True | True | True | True | True | False | False | False |
| Line Search Times | N/A | N/A | N/A | N/A | N/A | N/A | 25 | 25 | 25 |
| Line Search Decay | N/A | N/A | N/A | N/A | N/A | N/A | 0.8 | 0.8 | 0.8 |
| Proximal clip | 0.2 | 0.2 | 0.2 | 0.2 | 0.2 | 0.2 | N/A | N/A | N/A |
| Max horizon | 200 | 200 | 200 | 200 | 200 | 200 | 200 | 200 | 200 |

Table 3: Hyperparameters used for each baseline.

at https://www.gymlibrary.ml/. **Bullet Safety Gym**. The implementation of the Circle task comes from Bullet-safety-Gym (Gronauer 2022), which Stooke, Achiam, and Abbeel (2020) first proposed. The reward is dense and increases by the agent's velocity and the proximity to the boundary of the circle. Costs are received when the agent leaves the safety zone defined by the two yellow boundaries. The environment is open source at https://github.com/SvenGronauer/Bullet-Safety-Gym.

**Safety Gym**. The remaining two tasks, Goal and Button, are from Safety-Gym (Ray, Achiam, and Amodei 2019). Compared to Run and Circle tasks, they are more stochastic and sophisticated in that agents are challenged to maximize the return while satisfying the constraints.

The environment is open source at https://github.com/openai/safety-gym, and readers can see OpenAI's blog at https://openai.com/blog/safety-gym/ for more details.

## A.3 Agents

For single-constraint scenarios, Point agent is a 2D mass point($A \subseteq \mathbb{R}^2$) and Ant is an quadruped robot($A \subseteq \mathbb{R}^8$). For the multi-constraint scenario which is modified from OpenAI SafetyGym [30], $S \subseteq \mathbb{R}^{28+16\cdot m}$ where $m$ is the number of pseudo-radar (one for each type of obstacles and we set two different types of obstacles in the Navigation task) and $A \subseteq \mathbb{R}^2$ for a mass point or a wheeled car.

### A.3.1 Experimental Details

To be fair in comparison, the proposed e-COP algorithm and FOCOPS [43] are implemented with same rules and tricks on the code-base of [30].

### A.3.2 Hyperparameters

Table 3 shows the hyperparameters of baseline algorithms.

### A.3.3 Runtime Environment

All experiments were implemented in Pytorch 1.7 .0 with CUDA 11.0 and conducted on an Ubuntu 20.04.2 LTS with 8 CPU cores (AMD Ryzen Threadripper PRO 3975WX 8-Coresz), 127G memory and 2 GPU cards (NVIDIA GeForce RTX 4060 Ti Cards).

### A.3.4 Robustness to Cost Thresholds

We conducted a set of experiments wherein we study how e-COP effectively adapts to different cost thresholds. For this, we use a pre-trained e-COP agent, which is trained with a particular cost threshold in

an environment, and test its performance on different cost thresholds within the same environment. Figure 6 illustrates the training curves of these pre-trained agents, and we see that while e-COP can generalize well across different cost thresholds, other baseline algorithms may require further tuning to accommodate different constraint thresholds.

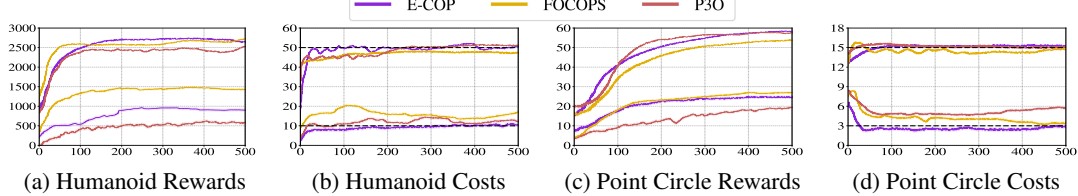

(a) Humanoid Rewards     (b) Humanoid Costs     (c) Point Circle Rewards     (d) Point Circle Costs

Figure 6: Cumulative episodic rewards and costs of baselines in two environments with two different constraint cost thresholds: 10 and 50 in Humanoid, and 3 and 15 in Point Circle. The hyperparameters are tuned at constraint limit of 20 in Humanoid and 10 in Point Circle.

