# OpenReview forum: "e-COP : Episodic Constrained Optimization of Policies"
_NeurIPS.cc/2024/Conference — NeurIPS 2024 poster_

### Official Review · Reviewer_p2dY · 2024-07-12

**Soundness:** 3
**Presentation:** 2
**Contribution:** 2
**Rating:** 5
**Confidence:** 5

**Summary:**

In this paper, the authors propose a policy optimization algorithm for constrained Reinforcement Learning (RL) problem in a finite horizon setting. First, the authors develop a policy difference lemma for the finite horizon MDP problem. Following that, the authors combine a set of ideas to propose the e-COP algorithm. The authors show that the proposed algorithm is numerically much stable compared to the state-of-the-art solutions. Optimality of the proposed scheme is established under certain assumptions. Extensive simulations have demonstrated that the performance of the algorithm is better than existing methods.

**Strengths:**

The paper is well-written and easy to follow. The results and claims presented in the paper appears to be correct.

**Weaknesses:**

The idea of using a quadratic penalty term to improve the stability near the edge of the constraint set boundary is a purely heuristic based approach. The idea of approximating $\rho_{\pi}$ by the empirical distribution from the policy of the previous episode does not appear to be a good approximation. The idea of allowing the agent to act in a constrained manner even before the constraint is violated may lead to suppression in the original objective function.

**Questions:**

1. The idea of using a quadratic penalty term to improve the stability near the edge of the constraint set boundary is a purely heuristic based approach. Is there any strong theoretical argument supporting such a choice?

2. In the related work section, the authors have discussed about drawbacks of using penalty terms in existing methods. Are these drawbacks applicable to the scheme proposed in the paper?

3. It is not clear how the time-dependent state occupation distribution is computed. The idea of approximating $\rho_{\pi}$ by the empirical distribution from the policy of the previous episode does not appear to be a good approximation. Although such approach may work well in infinite horizon problems where the optimality of stationary policy holds, it may not work well in the finite horizon setting.

4. The approach of using the second order penalty is not very novel and is widely used in practice for practical RL algorithms. What is the novelty of the proposed approach?

5. The idea of allowing the agent to act in a constrained manner even before the constraint is violated may lead to suppression in the original objective function. In other words, when the constraint is not too tight, the proposed scheme may unnecessarily impose additional level of constraint on the system.

6. How to ensure that we have sufficiently large $\beta$ and $\lambda_{t,i}$ which are well above the given lower bounds? The adaptive parameter selection process is just a heuristic.

7. The symbol in the last term of equation (7) is not clearly defined.

8. The algorithm proposed in the paper is an amalgamation of many ideas. Many of these ideas involve approximation and many are just driven from intuition without any formal guarantee of optimality. It is hard to see how this fusion work together in theory (e.g., satisfying the conditions given in Theorem 3.3).

**Limitations:**

Yes

---

> ### Author Rebuttal · Authors · 2024-08-05
>
> We thank the reviewer for comments on exposition and correctness. Responses below:
>
> We disagree with the reviewer on the weakness mentioned:
>
> (i) Calling introduction of quadratic penalty as purely heuristic minimizes the utility of such ideas that have proven to be central in making optimization algorithms practically useful. In optimization literature, it is a well known technique to improve numerical stability, and in fact our theoretical analysis for Theorem 3.3 accounts for it. See more under Q1.
>
> (ii) The idea of approximating $\rho_{\pi}$ by empirical distribution from the previous episode’s policy is again in the same spirit: Following a principled approach and yet introducing a series of good approximations that make the algorithm practical and yet, result in good or better performance. This is what has made PO algorithms so powerful that they are at the heart of most training of GenAI models (LLMs or Diffusion models) and find widespread usage on a daily basis.
>
> (iii) We have difficulty understanding the reviewer’s comment “ ... acting in a constrained manner even before the constraint is violated ... ”. Since the constraints depend on the entire episode, if you do not design policies that account for the constraints, then it may be too late to act to enforce them if you see the constraints are getting violated. Please do note that once the policy is designed as per our algorithm, there is no need for `constraint enforcement’. We have a guarantee (Theorem 3.3) that they will be satisfied.
>
> *Q1.* We respectfully, and strongly disagree that the introduction of quadratic penalty is purely heuristic. Our main result (text leading up to Theorem 3.3 and the theorem itself) talks about this very point. We theoretically prove that this attached quadratic penalty is an exact penalty from the primal problem in the constructed primal-dual problem. Thus, the constructed multiplier-penalty function is equivalent to the primal constrained problem to provide continuous and precise cost control. We mention in Lines 191-195 how the addition of the quadratic penalty theoretically helps the agent in more effective constraint control than its counterpart, the post-violation penalty approach.
>
> *Q2.* No they are not applicable. The only possible bottleneck in e-COP is the scalability, which can potentially be seen in some experiments. e-COP still promises to be a strong candidate for a safe RL baseline.
>
> *Q3.* You are correct in raising this concern. This has already been addressed in [1], which allows us to use such an approximation. Our extensive experiments on many environments showcase the ability of e-COP to outperform other SOTA baselines, so this approximation is even empirically proven to work.
>
> *Q4.* We do not use a second order penalty in our approach, we only deal with the expectation, which is first order. We have addressed the novelty of our work in Lines 57-68. Please see.
>
> *Q5.* To address this concern, we use an adaptive damping penalty that is instance dependent, adapts to the environment, and does not act “too safely”. This approach is discussed starting on Line 219, and ablation studies in Section 4.2 are also performed that show the effectiveness of our approach.
>
> *Q6.* For initial feasible parameters, we do a linesearch [2]. After initial feasible parameters, we update them as in Algorithm 2. Please go through the Appendix to see the theoretical justification, which we have provided in plenty. For your convenience, we provide it here as well:
>
> The gradient of $\mathcal{L}\_{t}(\mathbf{\pi}_{k}, \mathbf{\lambda}, \beta)$ takes the form
>
> $\nabla\_{\pi}\mathcal{L}\_{t}(\mathbf{\pi}\_{k},\mathbf{\lambda},\beta)=\nabla\_{\pi}\sum\_{h=t}^{H}\mathbb{E}\_{s\sim\rho\_{\pi\_{k,h}};a\sim\pi\_{k-1,h}}\big[-{\rho(\theta\_{h})}A^{\mathbf{\pi}\_{k-1}}\_{h}(s,a)\big]+\Sigma\_{\Psi\_{C\_{i},t}(\mathbf{\pi}\_{k-1},\mathbf{\pi}\_{k})\geq-\frac{\lambda\_{t,i}}{\beta}}\left(\lambda\_{t,i}+\beta\Psi\_{C_{i},t}(\mathbf{\pi}\_{k-1},\mathbf{\pi}\_{k})\right)\nabla\_{\pi}\Psi\_{C_{i},t}(\mathbf{\pi}\_{k-1},\mathbf{\pi}\_{k})$
>
> Suppose $(\pi^{\star},\mathbf{\lambda}^{\star})$ is the optimal policy and its corresponding dual parameters. Consider KKT condition to $(\pi^{\star},\mathbf{\lambda}^{\star})$ of the undamped problem (Eq. (5) with $\beta=0$):
>
> $\nabla\_{\pi}\mathcal{L}\_{t}(\pi^{\star},\mathbf{\lambda}^{\star})=\nabla\_{\pi}\sum\_{h=t}^{H}\mathbb{E}\_{s\sim\rho_{\pi_{k,h}};a\sim\pi_{k-1,h}}\big[-{\rho(\theta_{h})}A^{\mathbf{\pi}\_{k-1}}\_{h}(s,a)\big]+\Sigma\_{i}\lambda\_{t,i}^{\star}\nabla\_{\pi}\Psi\_{C_{i},t}(\mathbf{\pi}\_{k-1},\mathbf{\pi}^{\star})$.
>
> Since the above two equations are consistent for the optimal point $(\pi^{\star},\mathbf{\lambda}^{\star})$ as shown in Theorem 3.3, we can relate the constraint violation nearby this optimal point as $\max\left(\Psi\_{C_{i},t}(\mathbf{\pi}\_{k-1},\mathbf{\pi}\_{k-1}),-\frac{\lambda_{t,i}^{(k-1)}}{\beta^{(k-1)}}\right)\approx\frac{\lambda\_{t,i}^{\star}-\lambda\_{t,i}^{(k-1)}}{\beta^{(k-1)}}$.
>
> This means we can reduce the constraint violations of the policy iteration close to the optimal point by reasonably increasing $\beta$. Based on this, we remove the Lagrange dependency in Eq. (7) and postulate the adaptive parameter selection procedure.
>
> *Q7.* It is clearly defined, please see Line 181.
>
> *Q8.* Most if not all SOTA PO safe RL algorithms require approximations (please see the related works section). We don’t understand what you mean by “hard to see in theory”? We have provided theoretical justifications for the development of e-COP (Section 3) and our main result Theorem 3.3 exactly deals with the feasibility of our approach, which is extensively validated by our experiments.
>
> We hope these answers address your concerns, and really hope that you consider raising your score.
>
> [1] Bojun, Huang. "Steady state analysis of episodic reinforcement learning." NeurIPS 2020.
>
> [2] Achiam, Joshua, et al. "Constrained policy optimization." ICML 2017

---

> ### Comment · Reviewer_p2dY · 2024-08-10
>
> Thank you for your detailed response which has resulted in a better understanding of the paper. However, I still feel novelty of the paper is limited as it incorporates various ideas already existing in the literature, for a episodic setting.  I am happy to raise my score.

---

### Official Review · Reviewer_4GCx · 2024-07-12

**Soundness:** 3
**Presentation:** 2
**Contribution:** 2
**Rating:** 4
**Confidence:** 2

**Summary:**

The paper proposes a new algorithm for finite-horizon constrained RL problems. The algorithm is based on three ideas: PPO-like updates for the finite-horizon setting, P3O-like treatment of constraints with adaptive penalty coefficient, and quadratic damping penalty. The proposed algorithm outperformed previous algorithms in some benchmark tasks.

**Strengths:**

A strong empirical performance of the proposed algorithm. The introduction of quadratic damping penalty to safe RL seems to be a new idea. The paper is easy to follow except some typos.

**Weaknesses:**

Performance difference lemma for the finite-horizon setting is already known. See Kakade's thesis. Since PPO-like update is based on the performance difference lemma and PPO's idea, I do not see it is novel. P3O-like treatment of constraints with adaptive penalty coefficient is also not novel.

There are some parts I do not really understand. In particular I wonder if the proposed algorithm really does what Algorithm 2 states. Concretely Algorithm 2 suggests to save policy parameters for each time step, which sounds to be unusual. (Also I guess Line 6 of the algorithm has a type: $\pi_{k, t} \gets \pi_{k, t+1}$ should be $\pi_{k, t} \gets \pi_{k-1, t}$.)

If the algorithm is implemented as above, it is unfair to compare to previous algorithms, because previous algorithms use a much fewer number of parameters. If the algorithm is not implemented as above, there is a serious issue in presentation.

Eq 8 seems to contain many typos. For example $\min_{\pi_{k, t}} L(\theta, \lambda, \beta)$ does not make sense since the loss do not have $\pi_{k, t}$. (Sorry for the sloppy notation, but I believe the authors can understand what I mean.) Also $\pi_{k, t}^\star = \min_{\pi_{k, t}} L(\theta, \lambda, \beta)$ does not make sense since $\min_{\pi_{k, t}} L(\theta, \lambda, \beta)$ is a scalar.

**Questions:**

- Does the proposed algorithm what Algorithm 2 states? Algorithm 2 suggests to save policy parameters for each time step, which sounds to be unusual.
- I looked into the code, but it seems it will not run since `_episodic_reward_loss` and `_get_surr_cadv` methods  are undefined. I checked if the Github repo of omnisafe, but I cannot find those functions. Is the code intended only for reference? (Indeed the code is provided as a text file, not python file.)
- I currently think the main contribution of the paper is the introduction of quadratic penalty dumping to safe RL. Therefore I would like to see results of ablation studies to prove the importance of quadratic penalty dumping. Is there any result? (Or am I missing?)

**Limitations:**

If the proposed algorithm really does what Algorithm 2 states, there is an issue of huge memory requirement as the horizon gets larger. I think it should be discussed. Otherwise I do not see any potential negative societal impact.

---

> ### Author Rebuttal · Authors · 2024-08-05
>
> We thank the reviewer for comments on the novelty of the theory and strong empirical performance. Responses below:
>
> First please note that as far as we know this is the first policy optimization algorithm for the constrained or unconstrained setting as far as we know. Second, all policy optimization algorithms are based on a policy difference lemma, and so ours is not novel: we weren’t aware that for the episodic setting, it was already available in Kakade’s thesis. We will acknowledge and cite. We acknowledge that the penalty based constraints treatment is not novel. However, how to incorporate this treatment into episodic settings was unknown, as was how to update the policy network in the finite horizon setting.
>
> Note that we have a time-dependent policy (hence parameters are saved at each step). This is not a bug but a feature as stationary policies for the episodic setting are known to be sub-optimal. Line 6 of Algorithm 2 is correct, since we use the idea of backward recursion as is done in the classical dynamical programming algorithms. So, while it is true that our model stores a higher number of parameters, it is because this is needed for the episodic setting (stationary policies are suboptimal, nevertheless, if any one wishes to use our algorithm to find a stationary policy (with fewer parameters, it is quite straightforward to do so).
>
> The concern of fairness of comparison to these other algorithms designed for non-episodic settings is valid. Nevertheless, these are the closest baselines available and we show that their usage in the episodic setting will result in subpar performance (Also, we anticipated that reviewers will want to see comparison to such baselines even if they are not designed for this setting.). No other PO algorithms, with fewer or more parameters can perform as well as e-COP.
>
> The loss indeed has the minimizing variable, which is hidden in the importance sampling ratio as defined after Equation (4). We can make this distinction clear in the next draft. You are right in saying that the $\texttt{min}$ is incorrect, it is a typo, and it should be $\texttt{argmin}$, which we will also fix. Thank you for pointing this out.
>
> *Ques 1*: Yes, Algorithm 2 as stated is correct. Please see above.
>
> *Ques 2*: Yes, we had to code the two functions you mentioned, and we use commit SHA $\texttt{d55958a011df7800f256452e07811832cd2524d2}$ of omnisafe to run our experiments. The implementation of $\textunderscore get \textunderscore surr \textunderscore cadv$ function is based on the docs module in omnisafe.
>
> *Ques 3*: We have included ablation studies on the effect of the quadratic penalty in Section 4.2 with further details in Appendix A.3.4.
>
> We thank the reviewer for their review, and the many comments that have helped improve the paper substantially. If the reviewer is satisfied, we would really appreciate reconsideration of their current score.

---

> > ### Comment · Reviewer_4GCx · 2024-08-13
> >
> > Thank you very much for the rebuttal!
> >
> > > However, how to incorporate this treatment into episodic settings was unknown, as was how to update the policy network in the finite horizon setting.
> >
> > Would you explain this a bit more? I do not really see why converting algorithms for the infinite-horizon setting to the finite-horizon setting is untrivial. In my experience it is easy in theory, especially given performance difference lemma for the finite-horizon setting. For example which part of the derivation of e-COP is challenging in comparison to P3O?
> >
> > > Note that we have a time-dependent policy (hence parameters are saved at each step). This is not a bug but a feature as stationary policies for the episodic setting are known to be sub-optimal...
> >
> > Yes, I know in theory a non-stationary policy is required. However one can explicitly provide the remaining time to a policy network, as is done in Time Limits in Reinforcement Learning by Pardo et al.
> >
> > > Line 6 of Algorithm 2 is correct, since we use the idea of backward recursion as is done in the classical dynamical programming algorithms.
> >
> > Would you explain this point a bit more? Solving Eq 8 with a first-order optimizer as explained in Line 229-230 does not seem to result in backward recursion.
> >
> > > The concern of fairness of comparison to these other algorithms designed for non-episodic settings is valid... Nevertheless, these are the closest baselines available and we show that their usage in the episodic setting will result in subpar performance...
> >
> > But gradually removing some components from e-COP recover previous algorithms, no?
> >
> > > Ques 2: Yes, we had to code the two functions you mentioned, and we use commit SHA
> >
> > Your SHA is different from the one in readme.md of the supplementary material. Also would tell me where `_get_surr_cadv` is? I cannot find it. Maybe would you give me a github link to the file in which the method actually appears?
> >
> > > Ques 3: We have included ablation
> >
> > Thanks, but where is it? Maybe reviewers cannot see updated version yet...

---

> ### Author Response · Authors · 2024-08-13
>
> > Would you explain this a bit more? ...
>
> It is highly nontrivial due to the absence of a stationary state distribution. With a finite horizon, the time-dependent state occupation measure needs to be treated differently, as is done in e-COP. This different treatment requires completely new analysis to show that this approach is principled, and needs our main result (Theorem 3.3) to showcase that using occupation measure from the previous episode along with the quadratic penalty does not change the solution set of the original problem. This entire part is new and different when compared to P3O.
>
> > Yes, I know in theory a non-stationary policy is required ...
>
> There are two reasons why this approach is **theoretically** not correct (empirically maybe ok):
> 1. Augmenting a time dimension to the state space leads to sparse transitions, which causes the time-dependent state occupation measures to break down.
> 2. Time Limits in Reinforcement Learning paper deals with PPO in discounted episodic ($\gamma=0.99$) and un-discounted episodic ($\gamma=1$) setting. Comments: Firstly, for episodic RL, (un-discounted) cumulative episodic rewards are considered as metrics, rather than discounted rewards. Secondly, if you set $\gamma=1$, the policy difference lemma for discounted RL (as in Kakade's thesis) breaks down (see [1] for the solution to this problem). Due to this, no theoretical convergence or correctness statements can be made, which is why there are no theoretical justifications for such an approach.
>
> We, on the other hand, provide theoretical correctness proof and empirically showcase the effectiveness of our approach on various environments, showcasing the superior performance of e-COP compared to baselines.
>
> > Solving Eq 8 with a first-order optimizer ...
>
> See Lines 5-6 of Algorithm 2 for this. Eq. 8 is first-order solvable, and then backward recursion is done as part of the algorithm. From a careful look at Algorithm 2, it should be clear what is happening.
>
> > But gradually removing some components ...
>
> This is not correct: The introduction of quadratic penalty and backward recursion structure of the algorithm is not a linear addition, i.e., if one sets $\beta=0$ (so as to "remove" the quadratic penalty), one does not recover **any** previous algorithm. A novelty of our work lies in introducing the quadratic penalty for episodic settings and formalizing the policy update rule, all of which are independent of the baselines we have used.
>
> > Your SHA is different from the one ...
>
> We have **not** provided any readme.md with the supplementary material. Could you please tell which file are you referring to? For the code release, we are currently in the process of obtaining code release permission: We will make the code public once it is received, and before the paper is finalized.
>
> > Thanks, but where is it? Maybe reviewers cannot see updated version yet...
>
> The ablation studies are available in Section 4.2 ("Secondary Evaluation") of the submitted manuscript, with further details in Appendix A.3.4.
>
> We would also like to add that we very much appreciate the care and attention with which the reviewer has read the paper; the probing questions they ask will make our draft stronger ; and the willingness to engage with our response and cross-check our experimental data !
>
> [1] Agnihotri, A. et al. ACPO: A Policy Optimization Algorithm for Average MDPs with Constraints. ICML 2024.

---

### Official Review · Reviewer_kuft · 2024-07-13

**Soundness:** 3
**Presentation:** 3
**Contribution:** 3
**Rating:** 7
**Confidence:** 4

**Summary:**

The authors introduce a policy optimization algorithm for episodic constrained RL problems, e-COP. In general, Lagrangian formulation is used for constrained optimization problems, however, the constraints are not always satisfied in real world applications. The solution approach avoids Hessian matrix inversion. The algorithm approach uses deep learning-based function approximation, a KL divergence-based proximal trust region and gradient clipping popular in improving the generalization and avoiding vanishing gradients in proximal policy optimization algorithms. The authors use SafeGym to benchmark their algorithm against the state-of-the-art algorithms such as PPO, FOCOPS, CPO, PCPO, and P3O.

**Strengths:**

The e-COP algorithm in most cases outperforms all other baseline algorithms but fails on Humanoid, AndReach, and Grid where it provides the second best optimal results. The authors used an extensive set of baselines -- few using Lagrangian approximations.

The idea is original. In general, the paper is well-written and clear. There are few grammatical errors and typos.

**Weaknesses:**

The authors do not talk about the limitations of the algorithms. There is a mention on the complexity of the Grid environment resulting the e-COP not to perform as best. It will be good for the completion of the paper to have a short section on limitations.

**Questions:**

I would like to know whether any hyper-parameter tuning has been done for the baseline algorithms? It is possible that the Lagrangian formulation may perform better after tuning.

**Limitations:**

The authors do not adequately address limitations.

---

> ### Author Rebuttal · Authors · 2024-08-05
>
> We thank the reviewer for their comments on originality, and well-written exposition. We will fix any grammatical issues and typos. We do want to point out that while the algorithm does not perform the best on Humanoid, AntReach, and Grid, it still performs the second best on these three while performing the first best on the other five environments. Response to concerns below:
>
> With regards to the weaknesses, while there are no fundamental weaknesses, we shall include a section on limitations in the next version, wherein we will discuss the scalability issues we faced in the Grid environment. Since the application in mind for this paper is in diffusion models, we plan to conduct research and experiments in this direction to ascertain e-COP’s capabilities.
>
> For the baselines, some hyperparameter tuning has indeed been done, and we provide the hyperparameter values in the Appendix, which are set based on the best performing parameters for each algorithm. We refer to standard codebases that are widely used for benchmarking algorithms [1], [2]. Note that we have done similar hyperparameter tuning for the Lagrangian formulation, PPO-L (and the other algorithms) as well. The gains in performance are not because of hyperparameter tuning but from the fact that the constrained problem formulation is the right fit for the problem.
>
> We thank the reviewer for their review, and the many comments that have helped improve the paper substantially. If the reviewer is satisfied, we would appreciate reconsideration of their current score.
>
>
> [1] omnisafe: https://github.com/PKU-Alignment/omnisafe
>
> [2] FSRL: https://github.com/liuzuxin/FSRL/

---

> > ### Comment · Reviewer_kuft · 2024-08-14
> >
> > Thank you for the rebuttal comments. It will good to include in the paper that the baselines were implemented and optimized to the best of the author's abilities along with reference to the Appendix.

---

### Decision · Program_Chairs · 2024-09-25

**Decision:**

Accept (poster)

**Comment:**

The manuscript presents e-COP a novel policy optimization algorithm for constraint episodic Reinforcement Learning (RL). e-COP combines ideas from constrained numerical optimization and proposes a practical algorithm. Overall, the paper is well-written, extensive experiments are provided and the performance of e-COP is solid.

A few minor issues:
- The authors should discuss the limitations of their approach in more detail in the text
- A few words about fairness of comparisons in the text would be appreciated
- Releasing the source code is important

Despite the above minor issues and some novelty issues, most reviewers (and myself) believe that e-COP can be a valuable contribution to the RL community. As such I recommend "Accept (poster)".